# SPIDE: A Purely Spike-based Method for Training Feedback Spiking Neural Networks

## Abstract

Spiking neural networks (SNNs) with event-based computation are promising brain-inspired models for energy-efficient applications on neuromorphic hardware. However, most high-performance supervised SNN training methods in machine learning research, such as conversion from artificial neural networks or direct training with surrogate gradients, require complex computation not supported by spiking neurons, which hinders them from spike-based energy-efficient training. Among them, the recently proposed method, implicit differentiation on the equilibrium state (IDE), for training feedback SNNs is a promising way possible for generalization to spike-based learning with flexible network structures. In this paper, we study spike-based implicit differentiation on the equilibrium state (SPIDE) that extends IDE for supervised local learning with spikes, which could be possible for energy-efficient training on neuromorphic hardware. Specifically, we first introduce ternary spiking neuron couples to realize ternary outputs with the common neuron model, and prove that implicit differentiation can be solved by spikes based on this design. With this approach, the whole training procedure can be made as event-driven spike computation and weights are updated locally with two-stage average firing rates. Then to reduce the approximation error of spikes due to the finite simulation time steps, we propose to modify the resting membrane potential. Based on it, the average firing rate, when viewed as a stochastic estimator, achieves an unbiased estimation of iterative solution for implicit differentiation and the variance of this estimator is reduced. With these key components, we can train SNNs with either feedback or feedforward structures in a small number of time steps. Further, the firing sparsity during training demonstrates the great potential for energy efficiency. Meanwhile, even with these constraints, our trained models can still achieve competitive results on MNIST, CIFAR-10, CIFAR-100, and CIFAR10-DVS. Our proposed method demonstrates the great potential for energy-efficient training of SNNs on neuromorphic hardware.

## 1 Introduction

Spiking neural networks (SNNs) are brain-inspired models that transmit spikes between neurons for event-driven energy-efficient computation. SNNs can be implemented with less energy on neuromorphic hardware (Akopyan et al., 2015; Davies et al., 2018; Pei et al., 2019; Roy et al., 2019), which can remedy the defects of large energy consumption of artificial neural networks (ANNs).

Different from ANNs, however, directly supervised training of SNNs is a hard problem due to the complex spiking neuron model which is discontinuous. To solve this problem, converting ANNs to SNNs (Hunsberger & Eliasmith, 2015; Rueckauer et al., 2017; Sengupta et al., 2019; Rathi et al., 2019; Deng & Gu, 2021; Yan et al., 2021), or many other direct SNN training methods (Wu et al., 2018; Bellec et al., 2018; Jin et al., 2018; Shrestha & Orchard, 2018; Wu et al., 2019; Neftci et al., 2019; Zhang & Li, 2019; Kim et al., 2020; Zheng et al., 2021; Bohte et al., 2002; Zhang & Li, 2020; Kim et al., 2020; Xiao et al., 2021) have been proposed. While these methods could partly tackle the problems of unsatisfactory performance or high latency, they require complex computation for gradient calculation or approximation, which cannot be implemented by common spiking neurons on neuromorphic hardware. They aim at training SNNs on commonly used computational units, e.g. GPU, and deploying trained models for energy-efficient inference. However they do not consider if the training procedure could leverage the same spike-based computation for gradient calculation and training to reduce the large energy consumption during training as well.

Table 1: Comparison of different supervised SNN training methods with respect to performance, latency, structure flexibility, neuron model, spike-based or not, and neuromorphic plausibility.

| Method | High Perform. | Low Latency | Struc. Flexi. | Common Neuron Model | Spike-based | Neuro. Plaus. |
|---|---|---|---|---|---|---|
| ANN-SNN | ✓ | × | × | ✓ | × | Low |
| BPTT with Surrogate Gradients | ✓ | ✓ | ✓ | ✓ | × | Low |
| DFA with Spikes | × | ? | × | ✓ | ✓ | High |
| SpikeGrad (Thiele et al., 2019a) | ✓ | ? | × | × | ✓ | Medium |
| IDE (Xiao et al., 2021) | ✓ | ✓ | ✓ | ✓ | × | Low |
| **SPIDE (ours)** | ✓ | ✓ | ✓ | ✓ | ✓ | High |

A few previous works try to train SNNs with spikes (Guerguiev et al., 2017; Neftci et al., 2017; Samadi et al., 2017; O'Connor & Welling, 2016; Thiele et al., 2019b;a). They either are based on direct feedback alignment (DFA) (Nøkland, 2016) and performs poorly, or require impractical special neuron models (Thiele et al., 2019b;a). Besides, they only focus on feedforward network structures imitated from ANNs, which ignores feedback connections that are ubiquitous in the human brain and enable neural networks to be shallower and more efficient (Kubilius et al., 2019; Xiao et al., 2021). Actually, feedback structures suit SNNs more since SNNs will naturally compute with multiple time steps, which could reuse representations and avoid uneconomical costs to unfold along time that ANNs suffer from (Xiao et al., 2021). So training algorithms for feedback SNNs, which may also degrade to feedforward structures by taking feedback as zero, is worth more exploration.

An ideal SNN training method should tackle the common problems, be suitable for flexible structures (feedforward or feedback) and be spike-based with high neuromorphic plausibility. The implicit differentiation on the equilibrium state (IDE) method (Xiao et al., 2021), which is recently proposed to train feedback spiking neural networks (FSNNs), is a promising method that may generalize to spike-based learning for requirement. They derive that the forward computation of FSNNs converges to an equilibrium state, which follows a fixed-point equation. Based on it, they propose to train FSNNs by implicit differentiation on this equation, which tackles the common difficulties for SNN training including non-differentiability and large memory costs, and has interesting local update properties. In their method, however, they leverage general root-finding methods to solve implicit differentiation, which requires complex computation on standard computation systems.

In this work, we extend the IDE method to spike-based IDE (SPIDE), which fulfills our requirements and has great potential for energy-efficient training of SNNs on neuromorphic hardware, by introducing ternary spiking neuron couples and proposing to solve implicit differentiation by spikes based on them. Our method is also applicable to feedforward structures by degrading the feedback connection as zero. In practice, however, it may require long time steps to stabilize the training with spikes due to approximation error for gradients. So we further dive into the approximation error from the statistical perspective, and propose to simply adjust the resting potential of SNNs to achieve an unbiased estimation of gradients and reduce the estimation variance of SNN computation. With these methods, we can train our models in a small number of time steps, which could further improve the energy efficiency as well as the latency. Our contributions include:

1. We propose the SPIDE method that is the first to train high-performance SNNs by spikes with common neuron models. Specifically, we propose ternary spiking neuron couples and prove that implicit differentiation for gradient calculation can be solved by spikes based on this design. Our method is applicable to both feedback and feedforward structures.

2. We theoretically analyze the approximation error of solving implicit differentiation by spikes, and propose to modify the resting potential to remove the approximation bias and reduce the estimation variance, which enables training in a small number of time steps.

3. Experiments show the low latency and firing sparsity during training, which demonstrates the great potential for energy-efficient training of SNNs on neuromorphic hardware. The performance on MNIST, CIFAR-10, CIFAR-100 and CIFAR10-DVS are also competitive.

## 2 RELATED WORK

Early works seek biologically inspired methods to train SNNs, e.g. spike-time dependent plasticity (STDP) (Diehl & Cook, 2015) or reward-modulated STDP (Legenstein et al., 2008). Since the rise of successful ANNs, several works try to convert trained ANNs to SNNs to obtain high performance (Hunsberger & Eliasmith, 2015; Rueckauer et al., 2017; Sengupta et al., 2019; Rathi et al., 2019; Deng & Gu, 2021; Yan et al., 2021). However, they suffer from extremely large time steps

and their structures are limited in the scope of ANNs. Others try to directly train SNNs by imitating backpropagation throught time (BPTT) and use surrogate derivative for discontinuous spiking functions (Lee et al., 2016; Wu et al., 2018; Bellec et al., 2018; Jin et al., 2018; Shrestha & Orchard, 2018; Wu et al., 2019; Zhang & Li, 2019; Neftci et al., 2019; Zheng et al., 2021) or compute gradient with respect to spiking times (Bohte et al., 2002; Zhang & Li, 2020; Kim et al., 2020). However, they suffer from approximation error and large memory costs. Xiao et al. (2021) propose the IDE method to train feedback spiking neural networks, which decouples the forward and backward procedures and avoids the common SNN training problems. However, all these methods require complex computation during training rather than spike-based. A few works focusing on training SNNs with spikes either are based on feedback alignment and limited in simple datasets (Guerguiev et al., 2017; Neftci et al., 2017; Samadi et al., 2017; O'Connor & Welling, 2016), or require impractical special neuron models that require consideration of accumulated spikes for spike generation (Thiele et al., 2019b;a), which is impractical on neuromorphic hardware. And they are only applicable to feedforward architectures. Instead, we are the first to leverage spikes with common neuron models to train SNNs with feedback or feedforward structures. The comparison of different methods is illustrated in Table 1.

# 3 PRELIMINARIES

We first introduce preliminaries about spiking neurons and the IDE training method. The basic thought of IDE (Xiao et al., 2021) is to identify the underlying equilibrium states of FSNN computation so that gradients can be calculated based on implicit differentiation on the equilibrium state. We will briefly introduce the conclusion of equilibrium states in Section 3.2 and the IDE method in Section 3.3. For more descriptions about the background please refer to Appendix A.

## 3.1 SPIKING NEURAL NETWORK MODELS

Spiking neurons draw inspirations from the human brain to communicate with each other by spikes. Each neuron integrates information from input spike trains by maintaining a membrane potential through a differential equation, and generates an output spike once the membrane potential exceeds a threshold, following which the membrane potential is reset to the resting potential. We consider the commonly used integrate and fire (IF) model and simple current model, whose discretized computational form is:

$$\begin{cases} u_i\,[t+0.5] = u_i[t] + \sum_j w_{ij}s_j[t] + b, \\ s_i[t+1] = H(u_i\,[t+0.5] - V_{th}), \\ u_i[t+1] = u_i\,[t+0.5] - (V_{th} - u_{rest})s_i[t+1], \end{cases} \tag{1}$$

where $u_i[t]$ is the membrane potential of neuron $i$ at time step $t$, $s_i[t]$ is the binary output spike train of neuron $i$, $w_{ij}$ is the connection weight from neuron $j$ to neuron $i$, $b$ is bias, $H$ is the Heaviside step function, $V_{th}$ is the firing threshold, and $u_{rest}$ is the resting potential. We use subtraction as the reset operation. $u_{rest}$ is usually taken as 0 in previous work, while we will reconsider it in Section 4.3.

## 3.2 EQUILIBRIUM STATES OF FEEDBACK SPIKING NEURAL NETWORKS

Xiao et al. (2021) derive that the (weighted) average rate of spikes during FSNN computation with common neuron models would converge to an equilibrium state following a fixed-point equation given convergent inputs. We focus on the conclusions with the discrete IF model under both single-layer and multi-layer feedback structures. The single-layer structure has one hidden layer of neurons with feedback connections on this layer. The update equation of membrane potentials is:

$$\mathbf{u}[t+1] = \mathbf{u}[t] + \mathbf{W}\mathbf{s}[t] + \mathbf{F}\mathbf{x}[t] + \mathbf{b} - (V_{th} - u_{rest})\mathbf{s}[t+1], \tag{2}$$

where $\mathbf{u}[t]$ and $\mathbf{s}[t]$ are the vectors of membrane potentials and spikes of these neurons, $\mathbf{x}[t]$ is the input at time step $t$, $\mathbf{W}$ is the feedback weight matrix, and $\mathbf{F}$ is the weight matrix from inputs to these neurons. The average input and average firing rate are defined as $\overline{\mathbf{x}}[t] = \frac{1}{t+1}\sum_{\tau=0}^{t}\mathbf{x}[\tau]$ and $\boldsymbol{\alpha}[t] = \frac{1}{t}\sum_{\tau=1}^{t}\mathbf{s}[\tau]$, respectively. Define $\sigma(x) = \min(1, \max(0, x))$.

The equilibrium state of the single-layer FSNN is described as (Xiao et al., 2021): If the average inputs converge to an equilibrium point $\overline{\mathbf{x}}[t] \to \mathbf{x}^*$, and there exists $\gamma < 1$ such that $\|\mathbf{W}\|_2 \leq \gamma V_{th}$, then the average firing rates of FSNN with discrete IF model will converge to an equilibrium point $\boldsymbol{\alpha}[t] \to \boldsymbol{\alpha}^*$, which satisfies the fixed-point equation $\boldsymbol{\alpha}^* = \sigma\left(\frac{1}{V_{th}}\left(\mathbf{W}\boldsymbol{\alpha}^* + \mathbf{F}\mathbf{x}^* + \mathbf{b}\right)\right)$. Note that they take $u_{rest} = 0$ in this conclusion, if we consider nonzero $u_{rest}$, the constraint and the fixed-point equation should be $\|\mathbf{W}\|_2 \leq \gamma(V_{th} - u_{rest})$ and $\boldsymbol{\alpha}^* = \sigma\left(\frac{1}{V_{th} - u_{rest}}\left(\mathbf{W}\boldsymbol{\alpha}^* + \mathbf{F}\mathbf{x}^* + \mathbf{b}\right)\right)$.

The multi-layer structure incorporates more non-linearity into the equilibrium fixed-point equation, which has multiple layers with feedback connections from the last layer to the first layer. The update equations of membrane potentials are expressed as:

$$\begin{cases} \mathbf{u}^1[t+1] = \mathbf{u}^1[t] + \mathbf{W}^1\mathbf{s}^N[t] + \mathbf{F}^1\mathbf{x}[t] + \mathbf{b}^1 - (V_{th} - u_{rest})\mathbf{s}^1[t+1], \\ \mathbf{u}^l[t+1] = \mathbf{u}^l[t] + \mathbf{F}^l\mathbf{s}^{l-1}[t+1] + \mathbf{b}^l - (V_{th} - u_{rest})\mathbf{s}^l[t+1], \quad l = 2, \cdots, N. \end{cases} \tag{3}$$

The equilibrium state of the multi-layer FSNN with $u_{rest}$ is described as (Xiao et al., 2021): If the average inputs converge to an equilibrium point $\overline{\mathbf{x}}[t] \to \mathbf{x}^*$, and there exists $\gamma < 1$ such that $\|\mathbf{W}^1\|_2\|\mathbf{F}^N\|_2 \cdots \|\mathbf{F}^2\|_2 \leq \gamma(V_{th} - u_{rest})^N$, then the average firing rates of multi-layer FSNN with discrete IF model will converge to equilibrium points $\boldsymbol{\alpha}^l[t] \to \boldsymbol{\alpha}^{l^*}$, which satisfy the fixed-point equations $\boldsymbol{\alpha}^{1^*} = f_1\left(f_N \circ \cdots \circ f_2(\boldsymbol{\alpha}^{1^*}), \mathbf{x}^*\right)$ and $\boldsymbol{\alpha}^{l+1^*} = f_{l+1}(\boldsymbol{\alpha}^{l^*})$, where $f_1(\boldsymbol{\alpha}, \mathbf{x}) = \sigma\left(\frac{1}{V_{th}-u_{rest}}(\mathbf{W}^1\boldsymbol{\alpha} + \mathbf{F}^1\mathbf{x} + \mathbf{b}^1)\right)$ and $f_l(\boldsymbol{\alpha}) = \sigma\left(\frac{1}{V_{th}-u_{rest}}(\mathbf{F}^l\boldsymbol{\alpha} + \mathbf{b}^l)\right)$.

### 3.3 IDE TRAINING METHOD

Based on the equilibrium states in Section 3.2, we can train FSNNs by calculating gradients with implicit differentiation (Xiao et al., 2021). Let $\boldsymbol{\alpha} = f_\theta(\boldsymbol{\alpha})$ denote the fixed-point equation of the equilibrium state which is parameterized by $\theta$, $g_\theta(\boldsymbol{\alpha}) = f_\theta(\boldsymbol{\alpha}) - \boldsymbol{\alpha}$, and let $\mathcal{L}(\boldsymbol{\alpha}^*)$ denote the objective function with respect to the equilibrium state $\boldsymbol{\alpha}^*$. The implicit differentiation satisfies $\left(I - \frac{\partial f_\theta(\boldsymbol{\alpha}^*)}{\partial \boldsymbol{\alpha}^*}\right)\frac{\mathrm{d}\boldsymbol{\alpha}^*}{\mathrm{d}\theta} = \frac{\partial f_\theta(\boldsymbol{\alpha}^*)}{\partial\theta}$ (Bai et al., 2019) (we follow the numerator layout convention for derivatives). Therefore, the differentiation of $\mathcal{L}(\boldsymbol{\alpha}^*)$ for parameters can be calculated as:

$$\frac{\partial\mathcal{L}(\boldsymbol{\alpha}^*)}{\partial\theta} = -\frac{\partial\mathcal{L}(\boldsymbol{\alpha}^*)}{\partial\boldsymbol{\alpha}^*}\left(J_{g_\theta}^{-1}|_{\boldsymbol{\alpha}^*}\right)\frac{\partial f_\theta(\boldsymbol{\alpha}^*)}{\partial\theta}, \tag{4}$$

where $J_{g_\theta}^{-1}|_{\boldsymbol{\alpha}^*}$ is the inverse Jacobian of $g_\theta$ evaluated at $\boldsymbol{\alpha}^*$. The calculation of inverse Jacobian can be avoided by solving an alternative linear system (Bai et al., 2019; 2020; Xiao et al., 2021):

$$\left(J_{g_\theta}^\top|_{\boldsymbol{\alpha}^*}\right)\boldsymbol{\beta} + \left(\frac{\partial\mathcal{L}(\boldsymbol{\alpha}^*)}{\partial\boldsymbol{\alpha}^*}\right)^\top = 0. \tag{5}$$

Note that a readout layer after the last layer of neurons will be constructed for output (Xiao et al., 2021), which is equivalent to a linear transformation on the approximate equilibrium state, i.e. the output would be $\mathbf{o} = \mathbf{W}^o\boldsymbol{\alpha}^*$, and the loss will be calculated between $\mathbf{o}$ and labels $\mathbf{y}$ with a common criterion such as cross-entropy. Then the gradient on the equilibrium state could be calculated. For the solution of implicit differentiation, Xiao et al. (2021) follow Bai et al. (2019; 2020) to leverage root-finding methods, while we will solve it by spike dynamics, as will be derived in Section 4. We treat the forward computation of average firing rates $\boldsymbol{\alpha}[T]$ of FSNNs at time step $T$ roughly reach the equilibrium state. Then by substituting $\boldsymbol{\alpha}^*$ by $\boldsymbol{\alpha}[T]$ in the above equations, gradients for the parameters can be calculated only with $\boldsymbol{\alpha}[T]$ and the equation, and we calculate them based on spikes. With the gradients, first-order optimization methods such as SGD (Rumelhart et al., 1986) and its variants can be applied to update parameters.

## 4 SPIKE-BASED IMPLICIT DIFFERENTIATION ON THE EQUILIBRIUM STATE

In this section, we present our SPIDE method that calculates the whole training procedure based on spikes. We first introduce ternary spiking neuron couples in Section 4.1 and how to solve implicit differentiation in Section 4.2. Then we theoretically analyze the approximation error and propose the improvement in Section 4.3. Finally, a summary of the training pipeline is presented in Section 4.4.

### 4.1 TERNARY SPIKING NEURON COUPLES

The common spiking neuron model only generates spikes when the membrane potential exceeds a positive threshold, which limits the firing rate from representing negative information. To enable approximation of possible negative values for implicit differentiation calculation in Section 4.2, we require negative spikes, whose expression could be:

$$s_i[t+1] = T\left(u_i[t+0.5], V_{th}\right) = \begin{cases} 1, & u_i[t+0.5] > V_{th} \\ 0, & |u_i[t+0.5]| \leq V_{th} \\ -1, & u_i[t+0.5] < -V_{th} \end{cases}, \tag{6}$$

and the reset is the same as usual: $u_i[t+1] = u_i[t+0.5] - (V_{th} - u_{rest})s_i[t+1]$. Direct realization of such ternary output, however, may be not supported by common neuromorphic hardware for SNNs.

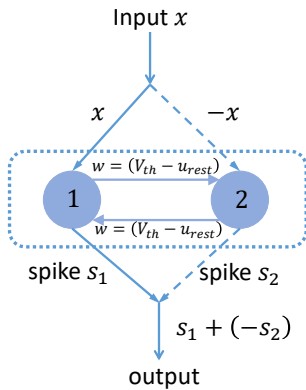

Figure 1: Illustration of ternary spiking neuron couples.

We propose to leverage two coupled common neurons to realize this computation. As illustrated in Figure 1, the two coupled neurons with the common IF model (Eq. (1)) receive opposite inputs and output opposite spikes, which aim to deal with positive information and negative information with spikes, respectively. They should share a reset operation in order to accord with Eq. (6), which can be realized by the connection between them: as we use subtraction as the reset operation, the connection whose weight equals $V_{th} - u_{rest}$ enables one neuron to reset another equivalently. To see how this works, consider the condition that the accumulated membrane potential of neuron 1 reaches $V_{th}$, then neuron 1 would generate a spike and reset, and the output is this positive spike. At the same time, the membrane potential of neuron 2 is $-V_{th}$ and the neuron will not fire and reset, but the spike from neuron 1 will reset it to $-u_{rest}$, which accords with our desired reset for ternary output. Similarly, if the inputs are negative, neuron 2 will generate a spike which will be treated negative as output, and both neurons are reset. For the operation of taking negative, one solution is to enable the reverse operation on hardware, another is to reconnect neuron 2 with other neurons while taking the weight negative to that of neuron 1. Therefore, such kind of coupled neurons with the common IF model could realize ternary output.

We note that the SpikeGrad algorithm (Thiele et al., 2019a) also requires neurons for ternary output. However, they do not consider how such kind of operation can be implemented with the common neuron model on neuromorphic hardware, and moreover, they propose another modified ternary model in practice that requires consideration of accumulated spikes for spike generation, which is further impractical on neuromorphic hardware. Differently, our method can be realized with the common neuron model suitable for neuromorphic hardware.

## 4.2 SOLVING IMPLICIT DIFFERENTIATION WITH SPIKES

Based on the coupled neurons in Section 4.1, we can solve implicit differentiation with spikes. For notation simplicity, we directly use Eq. (6) as a ternary neuron without detailing coupled neurons below. Our main focus is on solving Eq. (5) with spikes. The brief outline for the derivation is: we first derive the update equation of membrane potentials in SNN computation, then we derive the equivalent equation of the rate of spikes with eliminating perturbation, finally, we could prove that the rate of spikes converges to the solution of Eq. (5).

We first consider the single-layer condition. Let $\boldsymbol{\alpha}[T_F]$ denote the average firing rate of these neurons after the forward computation with time steps $T_F$ as an approximate equilibrium state (we treat the forward procedure as the first stage), $\mathbf{g} = \left(\frac{\partial \mathcal{L}}{\partial \boldsymbol{\alpha}[T_F]}\right)^{\top}$ denote the gradient of the loss function on this approximate equilibrium state, and $\mathbf{m} = \sigma'(\boldsymbol{\alpha}[T_F])$, $\mathbf{M} = \mathrm{Diag}(\mathbf{m})$ denote a mask indicator based on the firing condition in the first stage, where $\sigma'(x) = \begin{cases} 1, \ 0 < x < 1 \\ 0, \ \text{else} \end{cases}$. We will have another $T_B$ time steps in the second backward stage to calculate implicit differentiation. We set the input to these neurons as $\mathbf{g}$ at all time steps, which can be viewed as input currents (Zhang & Li, 2020; Xiao et al., 2021). Then along the inverse connections of neurons and with a mask on neurons or weights and an output rescale, the computation of FSNN with ternary neurons is calculated as:

$$\mathbf{u}[t+1] = \mathbf{u}[t] + \frac{1}{V_{th} - u_{rest}}(\mathbf{MW})^{\top}\mathbf{s}[t] + \mathbf{g} - (V_{th}^b - u_{rest}^b)\mathbf{s}[t+1], \qquad (7)$$

where $V_{th}, u_{rest}$ and $V_{th}^b, u_{rest}^b$ are the threshold and resting potential during the first and second stage, respectively. Define the 'average firing rate' at this second stage as $\boldsymbol{\beta}[t] = \frac{1}{t}\sum_{\tau=1}^{t}\mathbf{s}[\tau]$, and $\mathbf{u}[0] = \mathbf{0}, \mathbf{s}[0] = \mathbf{0}$, then through summation, we have:

$$\boldsymbol{\beta}[t+1] = \frac{1}{V_{th}^b - u_{rest}^b}\left(\frac{t}{t+1}\frac{1}{V_{th} - u_{rest}}(\mathbf{MW})^{\top}\boldsymbol{\beta}[t] + \mathbf{g} - \frac{\mathbf{u}[t+1]}{t+1}\right). \qquad (8)$$

Since there could be at most $t$ spikes during $t$ time steps, $\boldsymbol{\beta}$ would be bounded in the range of $[-1, 1]$. The membrane potential $\mathbf{u}_i[t]$ will maintain the exceeded terms, i.e. define $\mathbf{v}_i[t] = \left(\frac{t}{t+1}\frac{1}{V_{th} - u_{rest}}(\mathbf{MW})^{\top}\boldsymbol{\beta}[t] + \mathbf{g}\right)_i$, we can divide $\mathbf{u}_i[t]$ as $\mathbf{u}_i^E[t] + \mathbf{u}_i^B[t]$, where

$\mathbf{u}_i^E[t] = \max\left(\mathbf{v}_i[t] - V_{th}^b, 0\right) + \min\left(\mathbf{v}_i[t] + V_{th}^b, 0\right)$ is the exceeded term while $\mathbf{u}_i^B[t]$ is a bounded term (Xiao et al., 2021) which is typically bounded in the range of $[-V_{th}^b, V_{th}^b]$. Then, Eq. (8) turns into:

$$\boldsymbol{\beta}[t+1] = \phi\left(\frac{1}{V_{th}^b - u_{rest}^b}\left(\frac{t}{t+1}\frac{1}{V_{th} - u_{rest}}(\mathbf{MW})^\top\boldsymbol{\beta}[t] + \mathbf{g} - \frac{\mathbf{u}^B[t+1]}{t+1}\right)\right), \quad (9)$$

where $\phi(x) = \min(1, \max(-1, x))$. Note that if the input $\mathbf{g}$ and weight $(\mathbf{MW})^\top$ are in an appropriate range, there would be no exceeded term and therefore $\phi$ will not take effect. Indeed we will rescale the loss to control $\mathbf{g}$ in an appropriate range, as will be indicated in Section 4.3. With this consideration, we could derive that the average firing rate $\boldsymbol{\beta}[t]$ converges to the solution of Eq. (5).

**Theorem 1.** *If there exists $\gamma < 1$ such that $\|(\mathbf{MW})^\top\|_2 \leq \gamma(V_{th} - u_{rest})(V_{th}^b - u_{rest}^b)$, then the average firing rate $\boldsymbol{\beta}[t]$ will converge to an equilibrium point $\boldsymbol{\beta}[t] \to \boldsymbol{\beta}^*$. When $V_{th}^b - u_{rest}^b = 1$, and there exists $\lambda < 1$ such that $\|(\mathbf{MW})^\top\|_\infty \leq \lambda(V_{th} - u_{rest})$ and $\|\mathbf{g}\|_\infty \leq 1 - \lambda$, then $\boldsymbol{\beta}^*$ is the solution of Eq. (5).*

The proof and discussion of assumptions are in Appendix B. With Theorem 1, we can solve Eq. (5) by simulating this second stage of SNN computation to obtain the 'firing rate' $\boldsymbol{\beta}[T_B]$ as the approximate solution. Plugging this solution to Eq. (4), the gradients can be calculated by: $\nabla_\mathbf{W}\mathcal{L} = \frac{1}{V_{th} - u_{rest}}\mathbf{M}\boldsymbol{\beta}[T_B]\boldsymbol{\alpha}[T_F]^\top, \nabla_\mathbf{F}\mathcal{L} = \frac{1}{V_{th} - u_{rest}}\mathbf{M}\boldsymbol{\beta}[T_B]\overline{\mathbf{x}}[T_F]^\top, \nabla_\mathbf{b}\mathcal{L} = \frac{1}{V_{th} - u_{rest}}\mathbf{M}\boldsymbol{\beta}[T_B]$.

Note that in practice, even if the data distribution is not properly in the range of $\phi$, we can still view $\phi$ as a kind of clipping for improperly large numbers, which could be similar to empirical techniques like "gradient clipping" to stabilize the training.

Then we consider the extension to the multi-layer condition. Let $\boldsymbol{\alpha}^l[T_F], l = 1, 2, \cdots, N$ denote the average firing rate of neurons in layer $l$ after the forward computation, $\mathbf{g} = \left(\frac{\partial\mathcal{L}}{\partial\boldsymbol{\alpha}^N[T_F]}\right)^\top$ denote the gradient of the loss function on the approximate equilibrium state of the last layer, and $\mathbf{m}^l = \sigma'(\boldsymbol{\alpha}^l[T_F]), \mathbf{M}^l = \text{Diag}(\mathbf{m}^l)$ denote the mask indicators. Similarly, we will have another $T_B$ time steps in the second stage to calculate implicit differentiation. We set the input to the last layer as $\mathbf{g}$ at all time steps. Then along the inverse connections of neurons and with a mask on neurons or weights and an output rescale, the computation of FSNN with ternary neurons is calculated as:

$$\begin{cases} \mathbf{u}^N[t+1] = \mathbf{u}^N[t] + \frac{1}{V_{th} - u_{rest}}(\mathbf{M}^1\mathbf{W}^1)^\top\mathbf{s}^1[t] + \mathbf{g} - (V_{th}^b - u_{rest}^b)\mathbf{s}^N[t+1], \\ \mathbf{u}^l[t+1] = \mathbf{u}^l[t] + \frac{1}{V_{th} - u_{rest}}(\mathbf{M}^{l+1}\mathbf{F}^{l+1})^\top\mathbf{s}^{l+1}[t+1] - (V_{th}^b - u_{rest}^b)\mathbf{s}^l[t+1], \quad l = N-1, \cdots, 1. \end{cases}$$
$$(10)$$

The 'average firing rates' $\boldsymbol{\beta}^l[t]$ are similarly defined for each layer, and the equivalent form can be similarly derived as:

$$\begin{cases} \boldsymbol{\beta}^N[t+1] = \phi\left(\frac{1}{V_{th}^b - u_{rest}^b}\left(\frac{t}{t+1}\frac{1}{V_{th} - u_{rest}}(\mathbf{M}^1\mathbf{W}^1)^\top\boldsymbol{\beta}^1[t] + \mathbf{g} - \frac{\mathbf{u}^{N^B}[t+1]}{t+1}\right)\right), \\ \boldsymbol{\beta}^l[t+1] = \phi\left(\frac{1}{V_{th}^b - u_{rest}^b}\left(\frac{1}{V_{th} - u_{rest}}(\mathbf{M}^{l+1}\mathbf{F}^{l+1})^\top\boldsymbol{\beta}^{l+1}[t+1] - \frac{\mathbf{u}^{l^B}[t+1]}{t+1}\right)\right). \end{cases}$$
$$(11)$$

The convergence of the 'firing rate' at the last layer to the solution of Eq. (5) can be similarly derived as Theorem 1. However, we need to calculate gradients for each parameter as Eq. (4), which is more complex than the single layer condition. Actually, we can derive that the 'firing rates' at each layer converge to equilibrium points, based on which the gradients can be easily calculated with information from the adjacent layers. Theorem 2 gives a formal description.

**Theorem 2.** *If there exists $\gamma < 1$ such that $\|(\mathbf{M}^1\mathbf{W}^1)^\top\|_2\|(\mathbf{M}^N\mathbf{F}^N)^\top\|_2\cdots\|(\mathbf{M}^2\mathbf{F}^2)^\top\|_2 \leq \gamma(V_{th} - u_{rest})^N(V_{th}^b - u_{rest}^b)^N$, then the average firing rates $\boldsymbol{\beta}^l[t]$ will converge to equilibrium points $\boldsymbol{\beta}^l[t] \to \boldsymbol{\beta}^{l^*}$. When $V_{th}^b - u_{rest}^b = 1$, and there exists $\lambda < 1$ such that $\|(\mathbf{M}^1\mathbf{W}^1)^\top\|_\infty \leq \lambda(V_{th} - u_{rest}), \|(\mathbf{M}^l\mathbf{F}^l)^\top\|_\infty \leq \lambda(V_{th} - u_{rest}), l = 2, \cdots, N$ and $\|\mathbf{g}\|_\infty \leq 1 - \lambda^N$, then $\boldsymbol{\beta}^{N^*}$ is the solution of Eq. (5), and $\boldsymbol{\beta}^{l^*} = \left(\frac{\partial h_N(\boldsymbol{\alpha}^{N^*})}{\partial h_l(\boldsymbol{\alpha}^{N^*})}\right)^\top\boldsymbol{\beta}^{N^*}, l = N-1, \cdots, 1$, where $h_l(\boldsymbol{\alpha}^{N^*}) = f_l \circ \cdots \circ f_2\left(f_1(\boldsymbol{\alpha}^{N^*}, \mathbf{x}^*)\right), l = N, \cdots, 1.$*

The functions $f_l$ are defined in Section 3.2. For the proof please refer to Appendix C. With Theorem 2, by plugging the solutions explicitly into Eq. (4), the gradients can be calculated by $\nabla_{\mathbf{F}^l}\mathcal{L} = \frac{1}{V_{th}-u_{rest}}\mathbf{M}^l\boldsymbol{\beta}^l[T_B]\boldsymbol{\alpha}^{l-1}[T_F]^\top, l = 2, \cdots, N, \nabla_{\mathbf{F}^1}\mathcal{L} = \frac{1}{V_{th}-u_{rest}}\mathbf{M}^1\boldsymbol{\beta}^1[T_B]\bar{\mathbf{x}}[T_F]^\top, \nabla_{\mathbf{W}^1}\mathcal{L} = \frac{1}{V_{th}-u_{rest}}\mathbf{M}^l\boldsymbol{\beta}^l[T_B]\boldsymbol{\alpha}^N[T_F]^\top, \nabla_{\mathbf{b}^l}\mathcal{L} = \frac{1}{V_{th}-u_{rest}}\mathbf{M}^l\boldsymbol{\beta}^l[T_B]$.

Note that the gradient calculation shares an interesting local property, i.e. it is proportional to the firing rates of the two neurons connected by it: $\nabla_{\mathbf{F}_{i,j}^l}\mathcal{L} = \frac{1}{V_{th}-u_{rest}}\mathbf{m}_i^l\boldsymbol{\beta}_i^l\boldsymbol{\alpha}_j^{l-1}$. During calculation, since we will have the firing rate of the first stage before the second stage, this calculation can also be carried out by event-based calculation triggered by the spikes in the second stage. So it would be plausible on neuromorphic hardware as well.

Also, note that the theorems still hold if we degrade our feedback models to feedforward ones by setting feedback connections as zero. In this setting, the dynamics and equilibriums degrade to direct functional mappings, and the implicit differentiation degrades to the explicit gradient. We can still approximate gradients with this computation.

In the following, we take $V_{th}^b - u_{rest}^b = 1$ by default to fulfill the assumption of theorems (it may take other values if we correspondingly rescale the outputs and we set 1 for simplicity). Other techniques like dropout can also be integrated into the calculation. Please refer to Appendix D for details.

### 4.3 REDUCING APPROXIMATION ERROR

Section 4.2 derives that we can solve implicit differentiation with spikes, as the average firing rate will gradually converge to the solution. In practice, however, we will simulate SNNs for finite time steps, and a smaller number of time steps is better for lower energy consumption. This will introduce approximation error which may hamper training. In this subsection, we theoretically study the approximation error and propose to adjust the resting potential to reduce it. Inspired by the theoretical analysis on quantized gradients (Chen et al., 2020), we will analyze the error from the statistical perspective.

For the 'average firing rates' $\boldsymbol{\beta}^l[t]$ in Eq. (8) and the multilayer counterparts, the approximation error $e$ to the equilibrium states consists of three independent parts $e_e, e_r$ and $e_i$: the first is $\mathbf{u}^{l^E}[t+1]$ that is the exceeded term due to the limitation of spike number, the second is $\mathbf{u}^{l^B}[t+1]$ which can be viewed as a bounded random variable, and the third is the convergence error of the iterative update scheme without $\mathbf{u}^l[t+1]$, i.e. let $\mathbf{b}^l[t]$ denote the iterative sequences for solving $\boldsymbol{\beta}^{l^*}$ as $\mathbf{b}^l[t+1] = \frac{t}{t+1}\frac{1}{V_{th}-u_{rest}}(\mathbf{M}^{l+1}\mathbf{F}^{l+1})^\top\mathbf{b}^l[t]$, the convergence error is $\|\mathbf{b}^l[t]-\boldsymbol{\beta}^{l^*}\|$. The second part $e_r$ can be again decomposed into two independent components $e_r = e_q + e_s$: $e_q$ is the quantization effect due to the precision of firing rates ($\frac{1}{T}$ for $T$ time steps) if we first assume the same average inputs at all time steps, and $e_s$ is due to the random arrival of spikes rather than the average condition, as there might be unexpected output spikes, e.g. the average input is 0 and the expected output should be 0, but two large positive inputs followed by one larger negative input at the last time would generate two positive spikes while only one negative spike. So the error is divided into: $e = e_e + e_q + e_s + e_i$. Since the iterative formulation is certain for $e_i$, we focus on $e_e, e_q$ and $e_s$.

Firstly, the error $e_q$ due to the quantization effect is influenced by the input scale and time steps $T_B$. To enable proper input scale and smaller time steps, we will rescale the loss function by a factor $s_l$, since the magnitude of gradients considering the direct cross-entropy loss function is relatively small. We scale the loss to an appropriate range so that information can be propagated by SNNs in smaller time steps, and most signals are in the range of $\phi$ as analyzed in Section 4.2. The base learning rate will be scaled by $\frac{1}{s_l}$ correspondingly. This is also adopted by Thiele et al. (2019a).

Then given the scale and number of time steps, $e_q, e_e$ and $e_s$ can be treated as random variables from statistical perspective, and we view $\boldsymbol{\beta}^l[t]$ as stochastic estimators for the equilibrium states with $e_i$. For the stochastic optimization algorithms, the expectation and variance of the gradients are important for convergence and convergence rate (Bottou, 2010), i.e. we hope an unbiased estimation of gradients and smaller estimation variance. As for $e_e$ and $e_s$, they depends on the input data and the expectations are $\mathbb{E}[e_e] = 0, \mathbb{E}[e_s] = 0$ (the positive and negative parts have the same probability). While for $e_q$, it will depend on our hyperparameters $V_{th}^b$ and $u_{rest}^b$. Since the remaining terms in $\mathbf{u}^{l^B}[t+1]$ caused by the quantization effect is in the range of $[u_{rest}^b, V_{th}^b]$ for positive terms while $[-V_{th}^b, -u_{rest}^b]$ for negative ones, given $V_{th}^b - u_{rest}^b$ and considering the uniform distribution, only

when $u_{rest}^b = -V_{th}^b$, the expectation $\mathbb{E}[e_q] = 0$ for both positive and negative terms. Therefore, we should adjust the resting potential from commonly used 0 (Wu et al., 2018; Sengupta et al., 2019; Xiao et al., 2021) to $-V_{th}^b$ for unbiased estimation, as described in Proposition 1.

**Proposition 1.** *For fixed $V_{th}^b - u_{rest}^b$ and uniformly distributed inputs and $e_q$, only when $u_{rest}^b = -V_{th}^b$, $\boldsymbol{\beta}^l[t]$ are unbiased estimators for $\mathbf{b}^l[t]$.*

Also, taking $u_{rest}^b = -V_{th}^b$ achieves the smallest estimation variance for the quantization effect $e_q$, considering the uniform distribution on $[u_{rest}^b, V_{th}^b] \cup [-V_{th}^b, -u_{rest}^b]$. Since the effects of $e_e$, $e_s$ and $e_i$ are independent of $e_q$ and their variance is certain given inputs, it leads to Proposition 2.

**Proposition 2.** *Taking $u_{rest}^b = -V_{th}^b$ reduces the variance of estimators $\boldsymbol{\beta}^l[t]$.*

With this analysis, we will take $V_{th}^b = 0.5, u_{rest}^b = -0.5$ in the following to stabilize the training. For $V_{th}$ and $u_{rest}$ during the first forward stage, we will also take $u_{rest} = -V_{th}$.

### 4.4 Details and Training Pipeline

The original IDE method (Xiao et al., 2021) leverages other training techniques including modified batch normalization (BN) and restriction on weight spectral norm. Since the batch statistical information might be hard to obtain for calculation on neuromorphic hardware and we seek for algorithms that could be possible on it, we drop the BN component in our SPIDE method. The restriction on the weight norm, however, is necessary for the convergence of feedback models, as indicated in theorems. We will adjust it for a more friendly calculation, please refer to Appendix D for details.

We summarize our training pipeline as follows (we also provide detailed pseudocodes in Appendix E). There are two stages for forward and backward procedures respectively. In the first stage, SNNs will receive inputs and perform the calculation as Eq. (1,2,3) for $T_F$ time steps, after which we get the output from the readout layer, and save the average inputs as well as the average firing rates and masks of each layer for the second stage. In the second stage, the last layer of SNNs will receive gradients for outputs and perform calculation along the inverse connections as Eq. (6,7,10) for $T_B$ time steps, after which we get the 'average firing rates' of each layer. Based on the firing rates from two stages, the gradients for parameters can be calculated as in Section 4.2 and then the first-order optimization algorithm is applied to update the parameters.

## 5 Experiments

In this section, we conduct experiments to demonstrate the effectiveness of our method and the great potential for energy-efficient training. We simulate the computation on common computational units. Please refer to Appendix D for implementation details and descriptions.

We first evaluate the effectiveness of our method in a small number of time steps. As shown in Table 2, we can train high-performance models with low latency ($T_F = 30$) in a very small number time steps during training (e.g. $T_B = 50$), which indicates the low latency and high energy efficiency. Note that the ANN-SNN methods usually require hundreds to thousands of time steps just for satisfactory inference performance, and direct training methods show that relatively small time steps are enough for inference, while we are the first to demonstrate that even training of SNNs can be carried out with spikes in a very small number of time steps. This is due to our analysis and improvement to reduce the approximation error, as illustrated in the ablation study in Appendix F.1.

Then we analyze the firing rate statistics to demonstrate the potential of energy efficiency. Since the energy consumption on event-driven neuromorphic hardware is proportional to the number of spikes, we present the average firing rates for forward and backward stages (for backward, both positive and negative spikes are considered as firing) in Figure 2. It shows the firing sparsity of our method, and spikes are sparser in the backward stage with around only $3\%$. Combined with the small number of time steps, this demonstrates the great potential for the energy-efficient training of SNNs based on our method on neuromorphic hardware.

Finally we evaluate the performance of our method on MNIST (LeCun et al., 1998), CIFAR-10 and CIFAR-100 (Krizhevsky & Hinton, 2009). We compare our method to several ANN-SNN methods (Hunsberger & Eliasmith, 2015; Sengupta et al., 2019; Deng & Gu, 2021), direct SNN training methods (Wu et al., 2018; Xiao et al., 2021), and SpikeGrad (Thiele et al., 2019a). As shown in Table 3, we can train models with a small number of time steps and our trained models achieve

competitive results on MNIST and CIFAR-10. Compared with the original IDE method (Xiao et al., 2021), since we discard the BN component, our generalization performance is poorer (a detailed discussion is in Appendix F.2). Compared with SpikeGrad (Thiele et al., 2019a), we can use fewer neurons and parameters due to flexible network structure choices, and a small number of time steps while they do not report this important feature. Besides, we use common neuron models while they require impractical models, as indicated in Section 4.1. The results and discussion on CIFAR-100 and CIFAR10-DVS are in Appendix F.3 and F.4 due to the space limit, and our model could achieve 64.07% and 60.7% accuracy respectively. The result on CIFAR-100 is competitive for networks without BN, though it is poorer than IDE with BN. And the result on CIFAR10-DVS is competitive among results of common SNN models. It shows the effectiveness of our method even with constraints of purely spike-based training. Future work could seek normalization techniques friendly for neuromorphic computation and our desired algorithm to further improve the performance.

Table 2: Evaluation of training with different time steps in the backward stage. Training is on CIFAR-10 with AlexNet-F structure and $T_F = 30$. Results are based on 3 runs of experiments.

| $T_B$ | Mean±Std (Best) |
|---|---|
| 50 | 88.41%±0.48% (89.07%) |
| 100 | 89.17%±0.14% (89.35%) |
| 250 | 89.61%±0.11% (89.70%) |
| 500 | 89.57%±0.08% (89.67%) |

Figure 2: The average firing rates for forward and backward stages during training. 'A' means AlexNet-F, 'C' means CIFARNet-F, and $T$ means time steps for the backward stage.

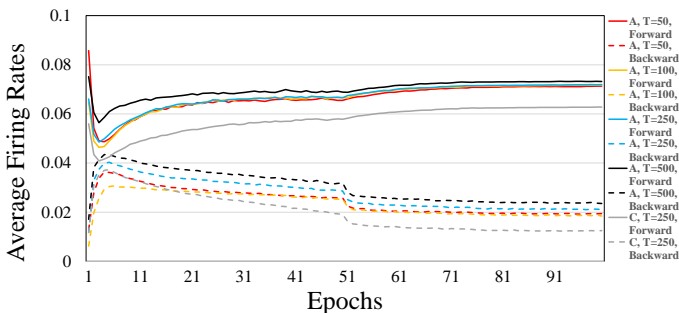

Table 3: Performance on MNIST and CIFAR-10. Results are based on 3 runs of experiments.

| MNIST | | | | | | |
|---|---|---|---|---|---|---|
| Method | Network structure | $T_F$ | $T_B$ | Mean±Std (Best) | Neurons | Params |
| BP (Lee et al., 2016) | 20C5-P2-50C5-P2-200 | >200 | / | (99.31%) | 33K | 518K |
| STBP (Wu et al., 2018) | 15C5-P2-40C5-P2-300 | 30 | / | (99.42%) | 26K | 607K |
| IDE (Xiao et al., 2021) | 64C5 (F64C5) | 30 | / | 99.53%±0.04% (99.59%) | 13K | 229K |
| SpikeGrad (Thiele et al., 2019a) | 15C5-P2-40C5-P2-300 | Unknown | Unknown | 99.38%±0.06% (99.52%) | 26K | 607K |
| **SPIDE (ours)** | 64C5s-64C5s-64C5 (F64C3u) | 30 | 100 | 99.34%±0.02% (99.37%) | 20K | 275K |
| **SPIDE (ours, degraded)** | 15C5-P2-40C5-P2-300 | 30 | 100 | 99.44%±0.02% (99.47%) | 26K | 607K |
| **CIFAR-10** | | | | | | |
| Method | Network structure | $T_F$ | $T_B$ | Mean±Std (Best) | Neurons | Params |
| ANN-SNN (Hunsberger & Eliasmith, 2015) | AlexNet | 80 | / | (83.52%) | 595K | 21M |
| ANN-SNN Sengupta et al. (2019) | VGG-16 | 2500 | / | (91.55%) | 311K | 15M |
| ANN-SNN (Deng & Gu, 2021) | CIFARNet | 400-600 | / | (90.61%) | 726K | 45M |
| STBP (Wu et al., 2019) | AlexNet | 12 | / | (85.24%) | 595K | 21M |
| STBP (w/o NeuNorm) (Wu et al., 2019) | CIFARNet | 12 | / | (89.83%) | 726K | 45M |
| STBP (Xiao et al., 2021) | AlexNet-F | 30 | / | (87.18%) | 159K | 3.7M |
| IDE (Xiao et al., 2021) | AlexNet-F | 30 | / | 91.74%±0.09% (91.92%) | 159K | 3.7M |
| IDE (Xiao et al., 2021) | CIFARNet-F | 30 | / | 92.08%±0.14% (92.23%) | 232K | 11.8M |
| SpikeGrad (Thiele et al., 2019a) | CIFARNet | Unknown | Unknown | 89.49%±0.28% (89.99%) | 726K | 45M |
| **SPIDE (ours)** | AlexNet-F | 30 | 250 | 89.61%±0.11% (89.70%) | 159K | 3.7M |
| **SPIDE (ours)** | CIFARNet-F | 30 | 250 | 89.94%±0.17% (90.13%) | 232K | 11.8M |

## 6 CONCLUSION

In this work, we propose the SPIDE method that generalize the IDE method to enable the whole training of SNNs with either feedback or degraded feedforward structures to be based on spikes with common neuron models. We prove that the implicit differentiation can be solved with spikes by our coupled neurons. We also analyze the approximation error due to finite time steps, and propose to adjust the resting potential of SNNs. Experiments show that we could achieve competitive performance with a small number of training time steps and sparse spikes, which demonstrates the great potential of our method for energy-efficient training of SNNs on neuromorphic hardware.

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

## A    MORE BACKGROUND ABOUT THE IDE TRAINING METHOD

Due to the complex spiking neuron model which is discontinuous, directly supervised training of SNNs is a hard problem, since the explicit computation is non-differentiable and therefore backpropagation along the forward computational graph could be problematic. The IDE training method (Xiao et al., 2021) considers another approach to calculating gradients that does not rely on the exact reverse of the forward computation, which avoids the problem of non-differentiability as well as large memory costs by BPTT-like methods with surrogate gradients. Specifically, the IDE training method first derives that the (weighted) average firing rate of FSNN computation with common neuron models would gradually evolve to an equilibrium state along time, which follows a fixed-point equation. Then by viewing the forward computation of FSNN as a black-box solver for this equation, and applying implicit differentiation on the equation, gradients can be calculated only based on this equation and the (weighted) average firing rate during forward computation rather than the exact forward procedure. Therefore, the forward and backward procedures are decoupled and the non-differentiability is avoided.

As briefly introduced in Section 3.2, the IDE method defines the average firing rates of spikes during forward computation, i.e. $\boldsymbol{\alpha}[t]$. Then IDE could derive an equivalent update equation for $\boldsymbol{\alpha}[t]$, based on the integrated update equations of membrane potentials Eq. (2,3). With the equivalent equation for $\boldsymbol{\alpha}[t]$, IDE proves that under certain conditions, $\boldsymbol{\alpha}[t]$ converges to an equilibrium state following a fixed-point equation. With the assumption that $\boldsymbol{\alpha}[T]$ after simulation of $T$ time steps roughly follows the fixed-point equation of the equilibrium state, gradients of the loss function for parameters can be calculated by implicit differentiation, as introduced in Section 3.3. So the training pipeline for the IDE method can be summarized as: first simulate FSNN computation for $T$ time steps to obtain the rate of spikes $\boldsymbol{\alpha}[T]$, then solve the implicit differentiation by root-finding methods for gradient calculation based on $\boldsymbol{\alpha}[T]$ and the derived fixed-point equation of equilibrium states, finally apply gradient-based optimizers to update parameters.

**Contribution of this work compared with IDE.**    Compared with IDE, this work extends the thought of equilibrium of spikes to solving implicit differentiation, which enables the whole training procedure to be based on spike computation with common neuron models and provides the potential for energy-efficient training of SNNs on neuromorphic hardware. IDE only derives the equilibrium states of forward FSNN computation and requires general root-finding methods with complex computation to solve implicit differentiation for training. It remains unclear if it could be solved by common spiking neuron models with only positive firing. This work design ternary spiking neuron couples and prove that the equilibrium of spike computation can be leveraged to solve implicit differentiation based on the design, and we propose to modify the resting membrane potential to make it practical in a relatively small number of time steps. This enables the proposed SPIDE method to be the first to train high-performance SNNs with low latency and firing sparsity by spikes with common neuron models, demonstrating the great potential for energy-efficient training of high-performance SNNs on neuromorphic hardware.

## B    PROOF OF THEOREM 1

*Proof.* We first prove the convergence of $\boldsymbol{\beta}[t]$. Let $V_u$ and $V_u^b$ denote $V_{th} - u_{rest}$ and $V_{th}^b - u_{rest}^b$ respectively. Consider $\|\boldsymbol{\beta}[t+1] - \boldsymbol{\beta}[t]\|$, it satisfies:

$$
\begin{aligned}
&\|\boldsymbol{\beta}[t+1] - \boldsymbol{\beta}[t]\| \\
=& \left\| \phi\left( \frac{1}{V_u^b}\left( \frac{t}{t+1}\frac{1}{V_u}(\mathbf{MW})^\top \boldsymbol{\beta}[t] + \mathbf{g} - \frac{\mathbf{u}^B[t+1]}{t+1} \right) \right) \right. \\
&\left. - \phi\left( \frac{1}{V_u^b}\left( \frac{t-1}{t}\frac{1}{V_u}(\mathbf{MW})^\top \boldsymbol{\beta}[t-1] + \mathbf{g} - \frac{\mathbf{u}^B[t]}{t} \right) \right) \right\| \\
\leq& \left\| \phi\left( \frac{1}{V_u^b}\left( \frac{1}{V_u}(\mathbf{MW})^\top \boldsymbol{\beta}[t] + \mathbf{g} \right) \right) - \phi\left( \frac{1}{V_u^b}\left( \frac{1}{V_u}(\mathbf{MW})^\top \boldsymbol{\beta}[t-1] + \mathbf{g} \right) \right) \right\| \\
&+ \left\| \phi\left( \frac{1}{V_u^b}\left( \frac{t}{t+1}\frac{1}{V_u}(\mathbf{MW})^\top \boldsymbol{\beta}[t] + \mathbf{g} - \frac{\mathbf{u}^B[t+1]}{t+1} \right) \right) - \phi\left( \frac{1}{V_u^b}\left( \frac{1}{V_u}(\mathbf{MW})^\top \boldsymbol{\beta}[t] + \mathbf{g} \right) \right) \right\|
\end{aligned}
$$

$$
\begin{aligned}
&+ \left\| \phi \left( \frac{1}{V_u^b} \left( \frac{t-1}{t} \frac{1}{V_u} (\mathbf{MW})^\top \boldsymbol{\beta}[t-1] + \mathbf{g} - \frac{\mathbf{u}^B[t]}{t} \right) \right) - \phi \left( \frac{1}{V_u^b} \left( \frac{1}{V_u} (\mathbf{MW})^\top \boldsymbol{\beta}[t-1] + \mathbf{g} \right) \right) \right\| \\
\leq \quad & \left\| \phi \left( \frac{1}{V_u^b} \left( \frac{1}{V_u} (\mathbf{MW})^\top \boldsymbol{\beta}[t] + \mathbf{g} \right) \right) - \phi \left( \frac{1}{V_u^b} \left( \frac{1}{V_u} (\mathbf{MW})^\top \boldsymbol{\beta}[t-1] + \mathbf{g} \right) \right) \right\| \\
&+ \frac{1}{V_u^b} \left( \left\| \frac{1}{t+1} \frac{1}{V_u} (\mathbf{MW})^\top \boldsymbol{\beta}[t] \right\| + \left\| \frac{\mathbf{u}^B[t+1]}{t+1} \right\| + \left\| \frac{1}{t} \frac{1}{V_u} (\mathbf{MW})^\top \boldsymbol{\beta}[t-1] \right\| + \left\| \frac{\mathbf{u}^B[t]}{t} \right\| \right).
\end{aligned}
\tag{12}
$$

As $\|(\mathbf{MW})^\top\|_2 \leq \gamma V_u V_u^b, \gamma < 1$, and $|\mathbf{u}_i^B[t]|$ is bounded, we have $\|\boldsymbol{\beta}[t+1]\| \leq \gamma \|\boldsymbol{\beta}[t]\| + \frac{1}{V_u} \left( \|\mathbf{g}\| + \left\| \frac{\mathbf{u}^B[t+1]}{t+1} \right\| \right) \leq \gamma \|\boldsymbol{\beta}[t]\| + c$, where $c$ is a constant. Therefore $\|\boldsymbol{\beta}[t]\|$ is bounded. Then $\forall \epsilon > 0, \exists T_1$ such that when $t > T_1$, we have:

$$
\frac{1}{V_u^b} \left( \left\| \frac{1}{t+1} \frac{1}{V_u} (\mathbf{MW})^\top \boldsymbol{\beta}[t] \right\| + \left\| \frac{\mathbf{u}^B[t+1]}{t+1} \right\| + \left\| \frac{1}{t} \frac{1}{V_u} (\mathbf{MW})^\top \boldsymbol{\beta}[t-1] \right\| + \left\| \frac{\mathbf{u}^B[t]}{t} \right\| \right) \leq \frac{\epsilon(1-\gamma)}{2}.
\tag{13}
$$

And since $\|(\mathbf{MW})^\top\|_2 \leq \gamma V_u V_u^b$, we have:

$$
\left\| \phi \left( \frac{1}{V_u^b} \left( \frac{1}{V_u} (\mathbf{MW})^\top \boldsymbol{\beta}[t] + \mathbf{g} \right) \right) - \phi \left( \frac{1}{V_u^b} \left( \frac{1}{V_u} (\mathbf{MW})^\top \boldsymbol{\beta}[t-1] + \mathbf{g} \right) \right) \right\| \leq \gamma \|\boldsymbol{\beta}[t] - \boldsymbol{\beta}[t-1]\|.
\tag{14}
$$

Therefore, when $t > T_1$, it holds that:

$$
\|\boldsymbol{\beta}[t+1] - \boldsymbol{\beta}[t]\| \leq \gamma \|\boldsymbol{\beta}[t] - \boldsymbol{\beta}[t-1]\| + \frac{\epsilon(1-\gamma)}{2}.
\tag{15}
$$

By iterating the above inequality, we have $\|\boldsymbol{\beta}[t+1] - \boldsymbol{\beta}[t]\| \leq \gamma^{t-T_1} \|\boldsymbol{\beta}[T_1+1] - \boldsymbol{\beta}[T_1]\| + \frac{\epsilon(1-\gamma)}{2} \left( 1 + \gamma + \cdots + \gamma^{t-T_1-1} \right) < \gamma^{t-T_1} \|\boldsymbol{\beta}[T_1+1] - \boldsymbol{\beta}[T_1]\| + \frac{\epsilon}{2}$. There exists $T_2$ such that when $t > T_1 + T_2$, $\gamma^{t-T_1} \|\boldsymbol{\beta}[T_1+1] - \boldsymbol{\beta}[T_1]\| \leq \frac{\epsilon}{2}$, and therefore $\|\boldsymbol{\beta}[t+1] - \boldsymbol{\beta}[t]\| < \epsilon$. According to Cauchy's convergence test, the sequence $\{\boldsymbol{\beta}[t]\}_{i=0}^\infty$ converges to $\boldsymbol{\beta}^*$. Considering the limit, it satisfies $\boldsymbol{\beta}^* = \phi \left( \frac{1}{V_u^b} \left( \frac{1}{V_u} (\mathbf{MW})^\top \boldsymbol{\beta}^* + \mathbf{g} \right) \right)$.

When $V_{th}^b - u_{rest}^b = 1$, and there exists $\lambda < 1$ such that $\|(\mathbf{MW})^\top\|_\infty \leq \lambda(V_{th} - u_{rest})$ and $\|\mathbf{g}\|_\infty \leq 1 - \lambda$, the equation turns into $\boldsymbol{\beta}^* = \phi \left( \frac{1}{V_u} (\mathbf{MW})^\top \boldsymbol{\beta}^* + \mathbf{g} \right)$. We have:

$$
\|\boldsymbol{\beta}^*\|_\infty = \left\| \phi \left( \frac{1}{V_u} (\mathbf{MW})^\top \boldsymbol{\beta}^* + \mathbf{g} \right) \right\|_\infty \leq \left\| \frac{1}{V_u} (\mathbf{MW})^\top \boldsymbol{\beta}^* \right\|_\infty + \|\mathbf{g}\|_\infty \leq \lambda \|\boldsymbol{\beta}^*\|_\infty + \|\mathbf{g}\|_\infty.
\tag{16}
$$

Therefore, $\|\boldsymbol{\beta}^*\|_\infty \leq \frac{\|\mathbf{g}\|_\infty}{1-\lambda} \leq 1$, and $\left\| \frac{1}{V_u} (\mathbf{MW})^\top \boldsymbol{\beta}^* + \mathbf{g} \right\|_\infty \leq \left\| \frac{1}{V_u} (\mathbf{MW})^\top \boldsymbol{\beta}^* \right\|_\infty + \|\mathbf{g}\|_\infty \leq \lambda + (1-\lambda) = 1$. It means $\boldsymbol{\beta}^* = \phi \left( \frac{1}{V_u} (\mathbf{MW})^\top \boldsymbol{\beta}^* + \mathbf{g} \right) = \frac{1}{V_u} (\mathbf{MW})^\top \boldsymbol{\beta}^* + \mathbf{g}$.

Taking $f_\theta(\boldsymbol{\alpha}^*) = \sigma \left( \frac{1}{V_{th} - u_{rest}} (\mathbf{W}\boldsymbol{\alpha}^* + \mathbf{F}\mathbf{x}^* + \mathbf{b}) \right)$ (i.e. the fixed-point equation at the equilibrium state as in Section 3.2) explicitly into Eq. (5), the linear equation turns into $\left( \frac{1}{V_u} (\mathbf{MW})^\top - I \right) \boldsymbol{\beta} + \mathbf{g} = 0$, where $V_u, \mathbf{M}, \mathbf{g}$ are previously defined. Therefore, $\boldsymbol{\beta}^*$ satisfies this equation. And since $\|(\mathbf{MW})^\top\|_2 \leq \gamma V_u, \gamma < 1$, the equation has the unique solution $\boldsymbol{\beta}^*$.

$\square$

**Remark 1.** *As for the assumptions in the theorem, firstly, when $V_{th}^b - u_{rest}^b = 1$ as we will take, the assumption for the convergence is weaker than that for the convergence in the forward stage (in Section 3.2), because $\|(\mathbf{MW})^\top\|_2 \leq \|\mathbf{W}\|_2$ as $\mathbf{M}$ is a diagonal mask matrix. We will restrict the*

*spectral norm of $\mathbf{W}$ following Xiao et al. (2021) to encourage the convergence of the forward stage (in Appendix D), then this backward stage would converge as well.*

*The assumptions for the consistency of the solution is a sufficient condition. In practice, the weight norm will be partially restricted by weight decay and our restriction on Frobenius norm (in Appendix D), as well as the diagonal mask matrix $\mathbf{M}$ which would be sparse if the forward firing events are sparse, and we will rescale the loss so that the input $\mathbf{g}$ is in an appropriate range, as indicated in Section 4.3. Even if these assumptions are not satisfied, we can view $\phi$ as a kind of empirical clipping techniques to stabilize the training, as indicated in Section 4.2. The discussion is similar for the multi-layer condition (Theorem 2) in the next section.*

## C  PROOF OF THEOREM 2

*Proof.* We first prove the convergence of $\boldsymbol{\beta}^l[t]$. Let $V_u$ and $V_u^b$ denote $V_{th} - u_{rest}$ and $V_{th}^b - u_{rest}^b$ respectively. Let $g_N^{t+1}(\boldsymbol{\beta}, \mathbf{g}, \mathbf{u}^B) = \phi\left(\frac{1}{V_u^b}\left(\frac{t}{t+1}\frac{1}{V_u}(\mathbf{M}^1\mathbf{W}^1)^\top\boldsymbol{\beta} + \mathbf{g} - \frac{\mathbf{u}^B}{t+1}\right)\right)$,

$g_l^t(\boldsymbol{\beta}, \mathbf{u}^B) = \phi\left(\frac{1}{V_u^b}\left(\frac{1}{V_u}(\mathbf{M}^{l+1}\mathbf{F}^{l+1})^\top\boldsymbol{\beta} - \frac{\mathbf{u}^B}{t}\right)\right), l = 1, \cdots, N-1,$

$g_N(\boldsymbol{\beta}, \mathbf{g}) = \phi\left(\frac{1}{V_u^b}(\frac{1}{V_u}(\mathbf{M}^1\mathbf{W}^1)^\top\boldsymbol{\beta} + \mathbf{g})\right),$

$g_l(\boldsymbol{\beta}) = \phi\left(\frac{1}{V_u^b}\left(\frac{1}{V_u}(\mathbf{M}^{l+1}\mathbf{F}^{l+1})^\top\boldsymbol{\beta}\right)\right), l = 1, \cdots, N-1.$

Then $\boldsymbol{\beta}^N[t+1] = g_N^{t+1}\left(g_1^t\left(\cdots g_{N-1}^t\left(\boldsymbol{\beta}^N[t], \mathbf{u}^{N-1B}[t]\right)\cdots, \mathbf{u}^{1B}[t]\right), \mathbf{g}, \mathbf{u}^{NB}[t+1]\right).$

We have:

$$
\begin{aligned}
&\left\|\boldsymbol{\beta}^N[t+1] - \boldsymbol{\beta}^N[t]\right\| \\
=\quad &\left\|g_N^{t+1}\left(g_1^t\left(\cdots g_{N-1}^t\left(\boldsymbol{\beta}^N[t], \mathbf{u}^{N-1B}[t]\right)\cdots, \mathbf{u}^{1B}[t]\right), \mathbf{g}, \mathbf{u}^{NB}[t+1]\right) \right.\\
&\left. -g_N^t\left(g_1^{t-1}\left(\cdots g_{N-1}^{t-1}\left(\boldsymbol{\beta}^N[t-1], \mathbf{u}^{N-1B}[t-1]\right)\cdots, \mathbf{u}^{1B}[t-1]\right), \mathbf{g}, \mathbf{u}^{NB}[t]\right)\right\| \\
\leq\quad &\left\|g_N\left(g_1\left(\cdots g_{N-1}\left(\boldsymbol{\beta}^N[t]\right)\cdots\right), \mathbf{g}\right) - g_N\left(g_1\left(\cdots g_{N-1}\left(\boldsymbol{\beta}^N[t-1]\right)\cdots\right), \mathbf{g}\right)\right\| \\
&+\left\|g_N^{t+1}\left(g_1^t\left(\cdots g_{N-1}^t\left(\boldsymbol{\beta}^N[t], \mathbf{u}^{N-1B}[t]\right)\cdots, \mathbf{u}^{1B}[t]\right), \mathbf{g}, \mathbf{u}^{NB}[t+1]\right) - g_N\left(g_1\left(\cdots g_{N-1}\left(\boldsymbol{\beta}^N[t]\right)\cdots\right), \mathbf{g}\right)\right\| \\
&+\left\|g_N^t\left(g_1^{t-1}\left(\cdots g_{N-1}^{t-1}\left(\boldsymbol{\beta}^N[t-1], \mathbf{u}^{N-1B}[t-1]\right)\cdots, \mathbf{u}^{1B}[t-1]\right), \mathbf{g}, \mathbf{u}^{NB}[t]\right) \right.\\
&\left. -g_N\left(g_1\left(\cdots g_{N-1}\left(\boldsymbol{\beta}^N[t-1]\right)\cdots\right), \mathbf{g}\right)\right\| \\
\leq\quad &\left\|g_N\left(g_1\left(\cdots g_{N-1}\left(\boldsymbol{\beta}^N[t]\right)\cdots\right), \mathbf{g}\right) - g_N\left(g_1\left(\cdots g_{N-1}\left(\boldsymbol{\beta}^N[t-1]\right)\cdots\right), \mathbf{g}\right)\right\| \\
&+\frac{1}{V_u^b}\left(\left\|\frac{1}{t+1}\frac{1}{V_u}(\mathbf{M}^1\mathbf{W}^1)^\top g_1^t\left(\cdots g_{N-1}^t\left(\boldsymbol{\beta}^N[t], \mathbf{u}^{N-1B}[t]\right)\cdots, \mathbf{u}^{1B}[t]\right)\right\|\right. \\
&+\underbrace{\left\|\frac{1}{V_u}(\mathbf{M}^1\mathbf{W}^1)^\top\left(g_1^t\left(\cdots g_{N-1}^t\left(\boldsymbol{\beta}^N[t], \mathbf{u}^{N-1B}[t]\right)\cdots, \mathbf{u}^{1B}[t]\right) - g_1\left(\cdots g_{N-1}\left(\boldsymbol{\beta}^N[t]\right)\cdots\right)\right)\right\|}_{A} \\
&+\left\|\frac{1}{t}\frac{1}{V_u}(\mathbf{M}^1\mathbf{W}^1)^\top g_1^{t-1}\left(\cdots g_{N-1}^{t-1}\left(\boldsymbol{\beta}^N[t-1], \mathbf{u}^{N-1B}[t-1]\right)\cdots, \mathbf{u}^{1B}[t-1]\right)\right\| \\
&+\underbrace{\left\|\frac{1}{V_u}(\mathbf{M}^1\mathbf{W}^1)^\top\left(g_1^{t-1}\left(\cdots g_{N-1}^{t-1}\left(\boldsymbol{\beta}^N[t-1], \mathbf{u}^{N-1B}[t-1]\right)\cdots, \mathbf{u}^{1B}[t-1]\right) - g_1\left(\cdots g_{N-1}\left(\boldsymbol{\beta}^N[t-1]\right)\cdots\right)\right)\right\|}_{B} \\
&+\left\|\frac{\mathbf{u}^{NB}[t+1]}{t+1}\right\| + \left\|\frac{\mathbf{u}^{NB}[t]}{t}\right\|\right).
\end{aligned}
\tag{17}
$$

For the term $A$ and $B$, they are bounded by:

$$
\begin{aligned}
A \leq \ & \frac{1}{V_u^b} \left( \left\| \frac{1}{V_u}(\mathbf{M}^1\mathbf{W}^1)^\top \frac{1}{V_u}(\mathbf{M}^N\mathbf{F}^N)^\top \left( g_2^t \left( \cdots g_{N-1}^t \left( \boldsymbol{\beta}^N[t], \mathbf{u}^{N-1\,B}[t] \right) \cdots, \mathbf{u}^{2\,B} \right) - g_2 \left( \cdots g_{N-1} \left( \boldsymbol{\beta}^N[t] \right) \cdots \right) \right) \right\| \right. \\
& \left. + \left\| \frac{1}{V_u}(\mathbf{M}^1\mathbf{W}^1)^\top \frac{\mathbf{u}^{1\,B}[t]}{t} \right\| \right) \\
\leq \ & \cdots\cdots \\
\leq \ & \frac{1}{V_u^b} \left\| \frac{1}{V_u}(\mathbf{M}^1\mathbf{W}^1)^\top \frac{\mathbf{u}^{1\,B}[t]}{t} \right\| + \cdots + \frac{1}{V_u^{b\,N-1}} \left\| \frac{1}{V_u^{N-1}}(\mathbf{M}^1\mathbf{W}^1)^\top (\mathbf{M}^N\mathbf{F}^N)^\top \cdots (\mathbf{M}^3\mathbf{F}^3)^\top \frac{\mathbf{u}^{N-1\,B}[t]}{t} \right\|,
\end{aligned}
\tag{18}
$$

and $B$ has the same form as $A$ by substituting $t$ with $t-1$.

Since $\|(\mathbf{M}^1\mathbf{W}^1)^\top\|_2 \|(\mathbf{M}^N\mathbf{F}^N)^\top\|_2 \cdots \|(\mathbf{M}^2\mathbf{F}^2)^\top\|_2 \leq \gamma V_u^N V_u^{b\,N}$, we have:

$$
\begin{aligned}
& \left\| g_N \left( g_1 \left( \cdots g_{N-1} \left( \boldsymbol{\beta}^N[t] \right) \cdots \right), \mathbf{g} \right) - g_N \left( g_1 \left( \cdots g_{N-1} \left( \boldsymbol{\beta}^N[t-1] \right) \cdots \right), \mathbf{g} \right) \right\| \\
\leq \ & \left\| \frac{1}{V_u^b} \frac{1}{V_u}(\mathbf{M}^1\mathbf{W}^1)^\top \left( g_1 \left( \cdots g_{N-1} \left( \boldsymbol{\beta}^N[t] \right) \cdots \right) - g_1 \left( \cdots g_{N-1} \left( \boldsymbol{\beta}^N[t-1] \right) \cdots \right) \right) \right\| \\
\leq \ & \cdots\cdots \\
\leq \ & \left\| \frac{1}{V_u^{b\,N}} \frac{1}{V_u^N}(\mathbf{M}^1\mathbf{W}^1)^\top (\mathbf{M}^N\mathbf{F}^N)^\top \cdots (\mathbf{M}^2\mathbf{F}^2)^\top \left( \boldsymbol{\beta}^N[t] - \boldsymbol{\beta}^N[t-1] \right) \right\| \\
\leq \ & \gamma \left\| \boldsymbol{\beta}^N[t] - \boldsymbol{\beta}^N[t-1] \right\|.
\end{aligned}
\tag{19}
$$

And since $\mathbf{u}_i^{l\,B}[t]$ is bounded, then $\forall \epsilon > 0, \exists T_1$ such that when $t > T_1$, we have:

$$
\left\| \boldsymbol{\beta}^N[t+1] - \boldsymbol{\beta}^N[t] \right\| \leq \gamma \left\| \boldsymbol{\beta}^N[t] - \boldsymbol{\beta}^N[t-1] \right\| + \frac{\epsilon(1-\gamma)}{2}.
\tag{20}
$$

Then $\|\boldsymbol{\beta}^N[t+1] - \boldsymbol{\beta}^N[t]\| < \gamma^{t-T_1}\|\boldsymbol{\beta}^N[T_1+1] - \boldsymbol{\beta}^N[T_1]\| + \frac{\epsilon}{2}$, and there exists $T_2$ such that when $t > T_1 + T_2$, $\|\boldsymbol{\beta}^N[t+1] - \boldsymbol{\beta}^N[t]\| < \epsilon$. According to Cauchy's convergence test, $\boldsymbol{\beta}^N[t]$ converges to $\boldsymbol{\beta}^{N^*}$, which satisfies $\boldsymbol{\beta}^{N^*} = g_N \left( g_1 \circ \cdots \circ g_{N-1}(\boldsymbol{\beta}^{N^*}), \mathbf{g} \right)$. Considering the limit, $\boldsymbol{\beta}^l[t]$ converges to $\boldsymbol{\beta}^{l^*}$, which satisfies $\boldsymbol{\beta}^{l^*} = g_l(\boldsymbol{\beta}^{l+1^*})$.

When $V_{th}^b - u_{rest}^b = 1$, and there exists $\lambda < 1$ such that $\|(\mathbf{M}^1\mathbf{W}^1)^\top\|_\infty \leq \lambda(V_{th} - u_{rest}), \|(\mathbf{M}^l\mathbf{F}^l)^\top\|_\infty \leq \lambda(V_{th} - u_{rest}), l = 2, \cdots, N$ and $\|\mathbf{g}\|_\infty \leq 1 - \lambda^N$, we have:

$$
\begin{aligned}
\left\| \boldsymbol{\beta}^{N^*} \right\|_\infty &= \left\| g_N \left( g_1 \circ \cdots \circ g_{N-1}(\boldsymbol{\beta}^{N^*}), \mathbf{g} \right) \right\|_\infty \leq \left\| \frac{1}{V_u}(\mathbf{M}^1\mathbf{W}^1)^\top g_1 \circ \cdots \circ g_{N-1}(\boldsymbol{\beta}^{N^*}) \right\|_\infty + \|\mathbf{g}\|_\infty \\
&\leq \lambda \left\| g_1 \circ \cdots \circ g_{N-1}(\boldsymbol{\beta}^{N^*}) \right\|_\infty + \|\mathbf{g}\|_\infty \leq \cdots\cdots \leq \lambda^N \left\| \boldsymbol{\beta}^{N^*} \right\|_\infty + \|\mathbf{g}\|_\infty
\end{aligned}
\tag{21}
$$

Therefore, $\left\| \boldsymbol{\beta}^{N^*} \right\|_\infty \leq \frac{\|\mathbf{g}\|_\infty}{1-\lambda^N} \leq 1$, and $\left\| \tilde{g}_{N-1}(\boldsymbol{\beta}^{N^*}) \right\|_\infty \leq \lambda \left\| \boldsymbol{\beta}^{N^*} \right\|_\infty \leq \lambda, \cdots\cdots,$ $\left\| \tilde{g}_1 \circ \cdots \circ \tilde{g}_{N-1}(\boldsymbol{\beta}^{N^*}) \right\|_\infty \leq \lambda^{N-1}, \left\| \tilde{g}_N \left( \tilde{g}_1 \circ \cdots \circ \tilde{g}_{N-1}(\boldsymbol{\beta}^{N^*}), \mathbf{g} \right) \right\|_\infty \leq \lambda^N + (1 - \lambda^N) = 1$, where $\tilde{g}_N(\boldsymbol{\beta}, \mathbf{g}) = \frac{1}{V_u}(\mathbf{M}^1\mathbf{W}^1)^\top \boldsymbol{\beta} + \mathbf{g}, \tilde{g}_l(\boldsymbol{\beta}) = \frac{1}{V_u}(\mathbf{M}^{l+1}\mathbf{F}^{l+1})^\top \boldsymbol{\beta}, l = 1, \cdots, N-1$, (i.e. $\tilde{g}_l$ is $g_l$ without the function $\phi$). It means $\boldsymbol{\beta}^{N^*} = g_N \left( g_1 \circ \cdots \circ g_{N-1}(\boldsymbol{\beta}^{N^*}), \mathbf{g} \right) = \tilde{g}_N \left( \tilde{g}_1 \circ \cdots \circ \tilde{g}_{N-1}(\boldsymbol{\beta}^{N^*}), \mathbf{g} \right)$ and $\boldsymbol{\beta}^{l^*} = g_l(\boldsymbol{\beta}^{l+1^*}) = \tilde{g}_l(\boldsymbol{\beta}^{l+1^*})$.

Taking $\boldsymbol{\alpha}^{1^*} = f_1 \left( f_N \circ \cdots \circ f_2(\boldsymbol{\alpha}^{1^*}), \mathbf{x}^* \right)$ and $\boldsymbol{\alpha}^{l+1^*} = f_{l+1}(\boldsymbol{\alpha}^{l^*})$ (i.e. the fixed-point equation at the equilibrium state as in Section 3.2) explicitly into Eq. (5), the linear equation turns into $\tilde{g}_1 \circ \cdots \circ \tilde{g}_{N-1}(\boldsymbol{\beta}) - \boldsymbol{\beta} + \mathbf{g} = 0$. Therefore, $\boldsymbol{\beta}^{N^*}$ satisfies this equation. And since $\|(\mathbf{M}^1\mathbf{W}^1)^\top\|_2 \|(\mathbf{M}^N\mathbf{F}^N)^\top\|_2 \cdots \|(\mathbf{M}^2\mathbf{F}^2)^\top\|_2 \leq \gamma V_u^N, \gamma < 1$, the equation has the unique

solution $\boldsymbol{\beta}^{N^*}$. Further, because $\tilde{g}_l(\boldsymbol{\beta}) = \left(\frac{\partial h_{l+1}(\boldsymbol{\alpha}^{N^*})}{\partial h_l(\boldsymbol{\alpha}^{N^*})}\right)^\top \boldsymbol{\beta}$, where $h_l(\boldsymbol{\alpha}^{N^*}) = f_l \circ \cdots \circ f_2\left(f_1(\boldsymbol{\alpha}^{N^*}, \mathbf{x}^*)\right), l = N, \cdots, 1$, we have $\boldsymbol{\beta}^{l^*} = \left(\frac{\partial h_N(\boldsymbol{\alpha}^{N^*})}{\partial h_l(\boldsymbol{\alpha}^{N^*})}\right)^\top \boldsymbol{\beta}^{N^*}, l = N-1, \cdots, 1$.

$\square$

# D  TRAINING DETAILS

## D.1  DROPOUT

Dropout is a commonly used technique to prevent over-fitting, and we follow Bai et al. (2019; 2020); Xiao et al. (2021) to leverage variational dropout, i.e. the dropout of each layer is the same at different time steps. Since applying dropout on the output of neurons is a linear operation with a mask and scaling factor, it can be integrated into the weight matrix without affecting the conclusions of convergence. The detailed computation with dropout is illustrated in the pseudocode in Appendix E.

## D.2  RESTRICTION ON WEIGHT NORM

As indicated in the theorems, a sufficient condition for the convergence to equilibrium states in both forward and backward stages is the restriction on the weight spectral norm. Xiao et al. (2021) leverages re-parameterization to restrict the spectral norm, i.e. they re-parameterize $\mathbf{W}$ as $\mathbf{W} = \alpha \frac{\mathbf{W}}{\|\mathbf{W}\|_2}$, where $\|\mathbf{W}\|_2$ is computed as the implementation of Spectral Normalization and $\alpha$ is a learnable parameter to be clipped in the range of $[-c, c]$ ($c$ is a constant). However, the computation of spectral norm and re-parameterization may be hard to realize on neuromorphic hardware. We adjust it for a more friendly calculation as follows.

First, the spectral norm is upper-bounded by the Frobenius norm: $\|\mathbf{W}\|_2 \leq \|\mathbf{W}\|_F$. We can alternatively restrict the Frobenius norm which is easier to compute. Further, considering that connection weights may not be easy for readout compared with neuron outputs, we can approximate $\|\mathbf{W}\|_F$ by $\|\mathbf{W}\|_F = \sqrt{\mathrm{tr}(\mathbf{W}\mathbf{W}^\top)} = \sqrt{\mathbb{E}_{\boldsymbol{\epsilon} \in \mathcal{N}(0, I_d)}\left[\|\boldsymbol{\epsilon}^\top \mathbf{W}\|_2^2\right]}$, according to the Hutchinson estimator (Hutchinson, 1989). It can be viewed as source neurons outputting noises and target neurons accumulating signals to estimate the Frobenius norm. We will estimate the norm based on the Monte-Carlo estimation (we will take 64 samples), which is similarly adopted by Bai et al. (2021) to estimate the norm of their Jacobian matrix. Then based on the estimation, we will restrict $\mathbf{W}$ as $\mathbf{W} = \alpha \frac{\mathbf{W}}{\|\mathbf{W}\|_F}$ where $\alpha = \min(c, \|\mathbf{W}\|_F)$, $c$ is a constant for norm range. This estimation and calculation may correspond to large amounts of noises in our brains, and a feedback inhibition on connection weights based on neuron outputs.

Following Xiao et al. (2021), we only restrict the norm of feedback connection weight $\mathbf{W}^1$ for the multi-layer structure, which works well in practice.

## D.3  OTHER DETAILS

For SNN models with feedback structure, we set $V_{th} = 1, u_{rest} = -1$ in the forward stage to form an equivalent equilibrium state as Xiao et al. (2021). The constant for restriction in Appendix D.2 is $c = 2$. Following Xiao et al. (2021), we train models by SGD with momentum for 100 epochs. The momentum is 0.9, the batch size is 128, and the initial learning rate is 0.05. For MNIST, the learning rate is decayed by 0.1 every 30 epochs, while for CIFAR-10 and CIFAR-100, it is decayed by 0.1 at the 50th and 75th epoch. We apply linear warmup for the learning rate in the first 400 iterations for CIFAR-10 and CIFAR-100. We apply the weight decay with $5 \times 10^{-4}$ and variational dropout with rate 0.2 for AlexNet-F and 0.25 for CIFARNet-F. The initialization of weights follows Wu et al. (2018), i.e. we sample weights from the standard uniform distribution and normalize them on each output dimension. The scale for the loss function (as in Section 4.3) is 100 for MNIST, 400 for CIFAR-10, and 500 for CIFAR-100.

For SNN models with degraded feedforward structure, our hyperparameters mostly follow Thiele et al. (2019a), i.e. we set $V_{th} = 0.5, u_{rest} = -0.5$, train models by SGD with momentum 0.9 for

60 epochs, set batch size as 128, and the initial learning rate as 0.1 which is decayed by 0.1 every 20 epochs, and apply the variational dropout only on the first fully-connected layer with rate 0.5.

The notations for our structures mean: '64C5' represents a convolution operation with 64 output channels and kernel size 5, 's' after '64C5' means convolution with stride 2 (which downscales $2\times$) while 'u' after that means a transposed convolution to upscale $2\times$, 'P2' means average pooling with size 2, and 'F' means feedback layers. The network structures for CIFAR-10 are:

AlexNet (Wu et al., 2019): 96C3-256C3-P2-384C3-P2-384C3-256C3-1024-1024,

AlexNet-F (Xiao et al., 2021): 96C3s-256C3-384C3s-384C3-256C3 (F96C3u),

CIFARNet (Wu et al., 2019): 128C3-256C3-P2-512C3-P2-1024C3-512C3-1024-512,

CIFARNet-F (Xiao et al., 2021): 128C3s-256C3-512C3s-1024C3-512C3 (F128C3u).

We simulate the computation on commonly used computational units. The code implementation is based on the PyTorch framework (Paszke et al., 2019), and experiments are carried out on one NVIDIA GeForce RTX 3090 GPU.

## E   Pseudocode of the SPIDE algorithm

Our algorithm consists of two-stage SNN computation, as explained in Section 4.4. The detailed computation for both stages are illustrated in Algorithm 1 and Algorithm 2 respectively.

---

**Algorithm 1** Forward procedure of SPIDE training - Stage 1.

---

**Input:** Network parameters $F^1, b^1, \cdots, F^N, b^N, W^1, W^o, b^o$; Input data $x$; Time steps $T_F$; Forward threshold $V_{th}$; Dropout rate $r$;
**Output:** Output of the readout layer $o$.

1: Initialize $u^i[0] = 0, i = 1, 2, \cdots, N$
2: If use dropout, randomly generate dropout masks $D^i(i = 1, 2, \cdots, N), D^f$ with rate $r$   // $D^i, D^f$ are saved for backward
3: **for** $t = 1, 2, \cdots, T_F$ **do**
4:    **if** $t == 1$ **then**
5:       $u^1[t] = u^1[t-1] + D^1 \odot (F^1 x[t] + b^1)$
6:    **else**
7:       $u^1[t] = u^1[t-1] + D^1 \odot (F^1 x[t] + b^1) + D^f \odot (W^1 s^N[t-1])$
8:    $s^1[t] = H(u^1[t] \geq V_{th})$
9:    $u^1[t] = u^1[t] - 2V_{th}s^1[t]$   // $u_{rest} = -V_{th}$, the same below
10:   **for** $l = 2, 3, \cdots, N$ **do**
11:      $u^l[t] = u^l[t-1] + D^l \odot (F^l s^{l-1}[t] + b^l)$
12:      $s^l[t] = H(u^l[t] \geq V_{th})$
13:      $u^l[t] = u^l[t] - 2V_{th}s^l[t]$
14:   $o = o + W^o s^N[t] + b^o$   // $o$ can be accumulated here, or calculated later by $o = W^o \alpha^N + b^o$
15: $o = \frac{o}{T}$
16: $\alpha^i = \frac{\sum_{t=1}^{T} s^i[t]}{T}, i = 1, 2, \cdots, N$   // Save for backward, firing rate in Stage 1
17: $m^i = H(\alpha^i > 0) \wedge H(\alpha^i < 1)$   // Save for backward, mask
18: If $x$ is not constant, save $x = \frac{\sum_{t=1}^{T} x[t]}{T}$ for backward
19: **return** $o$

---

## F   More experimental comparisons

### F.1   Ablation Study

In this section, we conduct ablation study on our improvement to reduce the approximation error by setting the resting potential as negative threshold. To formulate equivalent equilibrium states, we

---

**Algorithm 2** Backward procedure of SPIDE training - Stage 2.

---

**Input:** Network parameters $F^1, b^1, \cdots, F^N, b^N, W^1, W^o, b^o$; Forward output $o$; Label $y$; Time steps $T_B$; Forward threshold $V_{th}$; Backward threshold $V_{th}^b = 0.5$; Other hyperparameters and saved variables;

**Output:** Trained network parameters $F^1, b^1, \cdots, F^N, b^N, W^1, W^o, b^o$.

1: Calculate $g = \frac{\partial L(o,y)}{\partial o}$   // for CE loss, $\frac{\partial L(o,y)}{\partial o} = \text{softmax}(o) - y$, in practice we will scale the loss by a factor $s_l$, then $\frac{\partial L(o,y)}{\partial o} = s_l\left(\text{softmax}(o) - y\right)$

2: Initialize $u^i[0] = 0, i = 1, 2, \cdots, N$

3: **for** $t = 1, 2, \cdots, T_B$ **do**

4:    **if** $t == 1$ **then**

5:       $u^N[t] = u^N[t-1] + W^{o\top}g$

6:    **else**

7:       $u^N[t] = u^N[t-1] + W^{o\top}g + \frac{1}{2V_{th}}W^{1\top}(D^f \odot m^1 \odot s^1[t-1])$   // $m^i$ is the saved mask in Stage 1

8:    $s^N[t] = T(u^N[t], 0.5)$   // realized by two coupled neurons

9:    $u^N[t] = u^N[t] - s^N[t]$   // realized by two coupled neurons

10:    **for** $l = N-1, N-2, \cdots, 1$ **do**

11:       $u^l[t] = u^l[t-1] + \frac{1}{2V_{th}}F^{l\top}(D^l \odot m^{l+1} \odot s^{l+1}[t])$   // $m^i$ is the saved mask in Stage 1

12:       $s^l[t] = T(u^l[t], 0.5)$   // realized by two coupled neurons

13:       $u^l[t] = u^l[t] - s^l[t]$   // realized by two coupled neurons

14: $\beta^i = \frac{\sum_{t=1}^{T} s^i[t]}{T}, i = 1, 2, \cdots, N$   // "firing rate" in Stage 2

15: Calculate gradients:

16:    (1) $\nabla_{F^1}\mathcal{L} = \frac{1}{2V_{th}}(m^1 \odot \beta^1)x^\top$   // $m^i, x$ are the saved mask and average input in Stage 1

17:    (2) $\nabla_{F^i}\mathcal{L} = \frac{1}{2V_{th}}(m^i \odot \beta^i)\alpha^{i-1\top}, i = 2, 3, \cdots, N$   // $m^i, \alpha^i$ are the saved mask and firing rate in Stage 1

18:    (3) $\nabla_{b^i}\mathcal{L} = \frac{1}{2V_{th}}(m^i \odot \beta^i), i = 1, 2, \cdots, N$

19:    (4) $\nabla_{W^1}\mathcal{L} = \frac{1}{2V_{th}}(m^1 \odot \beta^1)\alpha^{N\top}$   // $m^i, \alpha^i$ are the saved mask and firing rate in Stage 1

20:    (5) $\nabla_{W^o}\mathcal{L} = \alpha^N\left(\frac{\partial L(o,y)}{\partial o}\right)^\top$   // $\alpha^i$ is the saved firing rate in Stage 1

21:    (6) $\nabla_{b^o}\mathcal{L} = \left(\frac{\partial L(o,y)}{\partial o}\right)^\top$

22: Update $F^1, b^1, \cdots, F^N, b^N, W^1, W^o, b^o$ based on the gradient-based optimizer   // SGD learning rate $\eta$ + momentum $\alpha$ & weight decay $\mu$, the base learning rate is scaled by the factor $s_l$ of the loss, i.e. $\eta = \frac{\eta}{s_l}$

23:    (1) Update the momentum $M_\theta = \alpha * M_\theta + (1 - \alpha) * \nabla_\theta\mathcal{L}, \theta \in \{F^i, b^i, W^1, W^o, b^o\}$

24:    (2) Update parameters $\theta = (1 - \mu) * \theta + \eta * M_\theta, \theta \in \{F^i, b^i, W^1, W^o, b^o\}$

25:    (3) Restrict the norm of $W^1$

26: **return** $F^1, b^1, \cdots, F^N, b^N, W^1, W^o, b^o$

---

take the same $V_{th} - u_{rest} = V_u$ and the same $V_{th}^b - u_{rest}^b = V_u^b$, and we consider the following settings: (1) both forward and backward stages apply our improvement, i.e. $u_{rest} = -V_{th}, u_{rest}^b = -V_{th}^b$; (2) remove the improvement on the backward stage, i.e. $V_{th}^b = V_u^b, u_{rest}^b = 0$; (3) remove the improvement on both forward and backward stages, i.e. $V_{th} = V_u, u_{rest} = 0$ and $V_{th}^b = V_u^b, u_{rest}^b = 0$. The latter two setting are denoted by "w/o B" and "w/o F&B" respectively.

The models are trained on CIFAR-10 with AlexNet-F structure and 30 forward time steps. The training and testing curves under different settings and backward time steps are illustrated in Figure 3 and Figure 4 respectively. It demonstrates that without our improvement, the training can not perform well within a small number of backward time steps, probably due to the bias and large variance of the estimated gradients. When the backward time steps are large, the performance gap is reduced since the bias of estimation is reduced. It shows the superiority of our improvement to training SNNs within a small number of backward time steps.

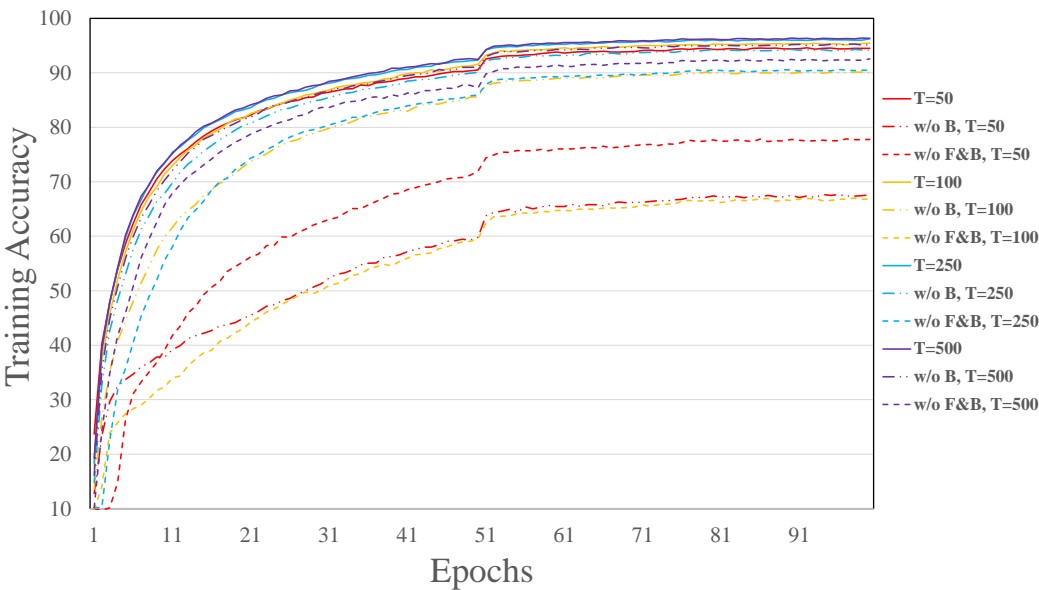

Figure 3: Comparison of training curves under different settings and backward time steps.

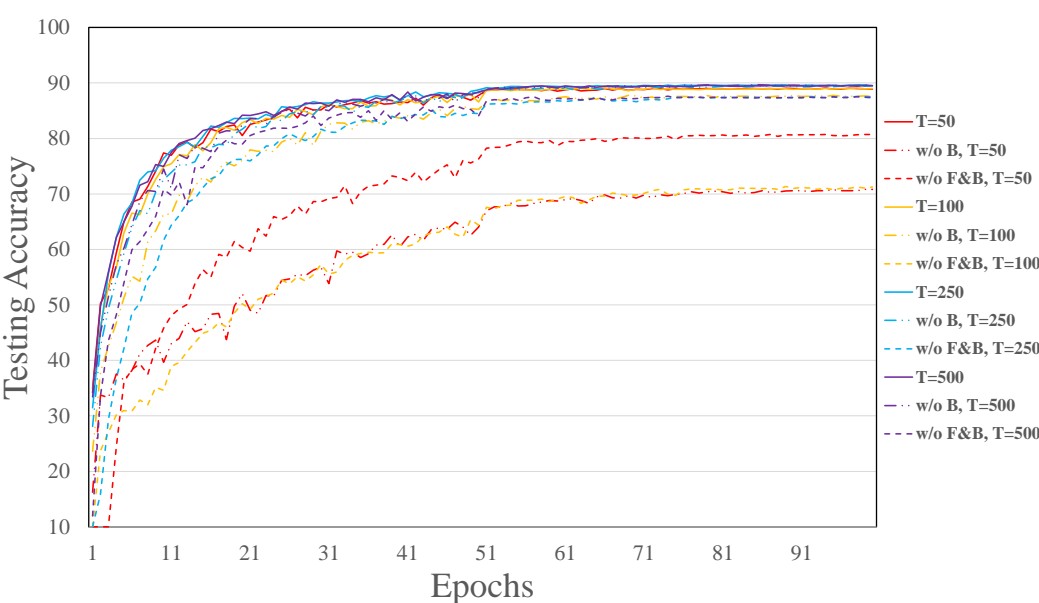

Figure 4: Comparison of testing curves under different settings and backward time steps.

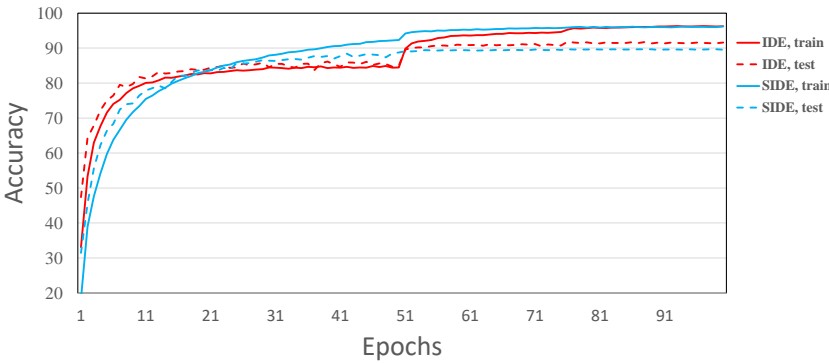

Figure 5: Comparison of training and testing curves between IDE and SPIDE on CIFAR-10 with AlexNet-F structure and $T_F = 30$.

## F.2 COMPARISON TO IDE

Table 3 in Section 5 shows that the SPIDE method performs poorer than the original IDE method (Xiao et al., 2021). We further investigate the training and testing curves during optimization to analyze this phenomenon. As shown in Figure 5, the SPIDE method could achieve the same training accuracy as the IDE method, while the generalization performance is poorer. Since the hyperparameters are the same for experiments, except that we drop the modified BN component (as explained in Section 4.4), the performance gap may be caused by the implicit regularization effect of BN. Therefore, the optimization ability of the SPIDE method should be similar to that of IDE, while future work would be to investigate how to realize operations similar to BN with possible computation that could be friendly on neuromorphic hardware.

Also, we note that models trained by the IDE method (Xiao et al., 2021) have sparser spikes, as their forward average firing rates are only around 1% (Xiao et al., 2021), while ours are around 7% as shown in Figure 2. It is again probably due to BN which would subtract the statistical mean value of neuron inputs, therefore regularizing the weights so that neurons will generate sparser spikes. An interesting future work is to further reduce the number of spikes considering this phenomenon.

## F.3 RESULTS ON CIFAR-100

Table 4: Performance on CIFAR-100. Results are based on 3 runs of experiments.

| Method | Network structure | BN | $T_F$ | $T_B$ | Mean±Std (Best) | Neurons | Params |
|---|---|---|---|---|---|---|---|
| BP (Thiele et al., 2019a) | CIFARNet | × | Unknown | / | (64.69%) | 726K | 45M |
| IDE (Xiao et al., 2021) | CIFARNet-F | ✓ | 30 | / | 71.56%±0.31% (72.10%) | 232K | 14.8M |
| SpikeGrad (Thiele et al., 2019a) | CIFARNet | × | Unknown | Unknown | (64.40%) | 726K | 45M |
| **SPIDE (ours)** | CIFARNet-F | × | 30 | 100 | 63.57%±0.30%(63.91%) | 232K | 14.8M |
| **SPIDE (ours)** | CIFARNet-F | × | 30 | 250 | 64.00%±0.11%(64.07%) | 232K | 14.8M |

In this section, we present the results on CIFAR-100. As shown in Table 4, our model could achieve 64.07% accuracy. Compared with IDE, the performance is poorer, and the main reason is probably again the absence of BN which could be important for alleviating overfitting on CIFAR-100 with relatively small number of images per class. The training accuracy of SPIDE is similar to IDE (around 93% v.s. around 94%) while the generalization performance is poorer. Despite this, the performance of our model is competitive for networks without BN and our model is with fewer neurons and parameters and a small number of time steps. Compared with SpikeGrad (Thiele et al., 2019a), we can use fewer neurons and parameters due to flexible network structure choices, and we leverage common neuron models while they require complex impractical models. Future work could investigate more suitable structures and if there are normalization techniques friendly for neuromorphic computation and our desired algorithm to further improve the performance.

## F.4 RESULTS ON CIFAR10-DVS

In this section, we supplement some results on the spiking dataset CIFAR10-DVS (Li et al., 2017). The CIFAR10-DVS dataset is the neuromorphic version of the CIFAR-10 dataset converted by a Dynamic Vision Sensor (DVS), which is composed of 10,000 samples, one-sixth of the original CIFAR-10. It consists of spike trains with two channels corresponding to ON- and OFF-event spikes. The pixel dimension is expanded to $128 \times 128$. Following the common practice, we split the dataset into 9000 training samples and 1000 testing samples. As for the data pre-processing, we reduce the time resolution by accumulating the spike events (Fang et al., 2021) into 30 time steps, and we reduce the spatial resolution into $48 \times 48$ by interpolation. We apply the random crop augmentation as CIFAR-10 to the input data. We leverage the network structure: 512C9s (F512C5), where the notations follow Appendix D.3. We train the model by SGD with momentum for 70 epochs. The momentum is 0.9, the batch size is 128, the weight-decay is $5 \times 10^{-4}$, and the initial learning rate is 0.05 which is decayed by 0.1 at the 50th epoch. No dropout is applied. The initialization of weights follows the widely used Kaiming initialization. The constant for restriction in Appendix D.2 is $c = 10$ due to the large channel size, and the scale for the loss function as well as the firing thresholds and resting potentials are the same as the CIFAR-10 experiment.

Table 5: Performance on CIFAR10-DVS.

| Method | Model | $T_F$ | $T_B$ | Accuracy |
|---|---|---|---|---|
| Gabor-SNN (Sironi et al., 2018) | Gabor-SNN | / | / | 24.5% |
| HATS (Sironi et al., 2018) | HATS | / | / | 52.4% |
| STBP (Wu et al., 2019) | Spiking CNN (LIF, w/o NeuNorm) | 40 | / | 58.1% |
| STBP (Wu et al., 2019) | Spiking CNN (LIF, w/ NeuNorm) | 40 | / | 60.5% |
| Tandem Learning (Wu et al., 2021) | Spiking CNN (IF) | 20 | / | 58.65% |
| Spike-based BP (Fang et al., 2021) | Spiking CNN (PLIF, w/ BN) | 20 | / | 74.8% |
| **SPIDE (ours)** | Spiking CNN (IF) | 30 | 250 | 60.7% |

As shown in Table 5, our model could achieve 60.7% accuracy, which is competitive among results of common SNN models, demonstrating the effectiveness of our method. Fang et al. (2021) leverages many techniques such as learnable membrane time constant, batch normalization, and max pooling to achieve better performance. We do not aim at outperforming the state-of-the-art results, but demonstrate that a competitive performance could be achieved even with our constraints of purely spike-based training in a relatively small number of time steps, verifying the effectiveness.

