# OpenReview forum: "SPIDE: A Purely Spike-based Method for Training Feedback Spiking Neural Networks"
_ICLR.cc/2022/Conference — ICLR 2022 Submitted_

### Official Review · Reviewer_PJRp · 2021-11-01

**Correctness:** 3
**Technical Novelty And Significance:** 3
**Empirical Novelty And Significance:** 3
**Recommendation:** 5
**Confidence:** 3

**Main Review:**

If one has an application of this method for neuromorphic hardware in mind, it becomes essential that one does not arrive in a rate-based coding regime. It has been demonstrated, e.g. for Intel's Loihi chip in (Davies et al., 2021) that one hardly gets and energy advantage in spike-based hardware if one works in a rate-based coding regime. Hence I am missing a discussion of this issue, and methods for arriving at sparsely firing neurons. If one aims instead at biologically plausible models, one also would have to look for solutions with biologically realistic firing rates of a few Hz.

Altogether I find it difficult to identify the specific innovations of this paper because of its inadequate literature review. In particular, there are already quite a number of spiking neural network solutions that achieve comparable or higher accuracies with comparable latency and number of parameters, see e.g.

Zhou S, Li X, Chen Y, et al. Temporal-Coded Deep Spiking Neural Network with Easy Training and Robust Performance[C]//Proceedings of the AAAI Conference on Artificial Intelligence. 2021, 35(12): 11143-11151.
Fang W, Yu Z, Chen Y, et al. Incorporating learnable membrane time constant to enhance learning of spiking neural networks[C]//Proceedings of the IEEE/CVF International Conference on Computer Vision. 2021: 2661-2671.

The abstract makes the really imprecise statement "--most SNN training methods require complex computation or impractical neuron models", which are hard to reconcile with publications such as

Bellec, G., Scherr, F., Subramoney, A., Hajek, E., Salaj, D., Legenstein, R., & Maass, W. (2020). A solution to the learning dilemma for recurrent networks of spiking neurons. Nature communications, 11(1), 1-15.

The authors should summarize and discuss more benefits of the proposed method. It is important for enhancing the impact of this work and drawing more audiences in the neuromorphic community.  In section 4.1, please explain why negative information is necessary for implicit differentiation calculation.




**Summary Of The Paper:**

The paper aims at porting the IDE method  into a spike-fbased and more bio-plausible version. The previous IDE used firing rates rather than spikes for computation, although reference Xiao et al in NeurIPS 2021 had already addressed implementations in spiking neural networks. The authors analyze the approximation error that results from solving implicit differentiation by spikes and report a solution based on ternary spiking neurons, that can be implemented with pairs of standard spiking neurons. They achieve in this way quite good performance for MNIST and CIFAR10.

**Summary Of The Review:**

The paper presents a nice step in an interesting direction. But it does not manage to clarify what exactly its innovations and possible applications are.

---

> ### Author Response · Authors · 2021-11-17
> **Response to Review PJRp (1/2)**
>
> Thank you very much for your valuable comments. We try our best to address your concerns as follows.
>
> 1. **First, we supplement some discussion about rate-based coding.**
>
> We respectfully disagree that “it becomes essential that one does not arrive in a rate-based coding regime”. The reason that rate-based coding is often regarded as impractical is because conventional methods to directly convert ANNs to SNNs requires large time steps and large firing rate, which might hardly get energy advantage since the energy consumption is proportional to the spike number. So many recent works regarding direct SNN training (most of which are also essentially rate-based coding) are trying to solve this problem by reducing the time step, and some could demonstrate very sparse firing so that the spike number is significantly reduced. For example, in the results of IDE (Xiao et al., NeurIPS 2021), the average firing rate is only around 0.7%, and given 30 time steps, each neuron only generates 0.2 spikes on average. We appreciate other coding schemes such as temporal coding, which demonstrate the potential for sparse spikes from another prospective direction. However, temporal coding may need special neuron models [1,2] or special circuits design [3], which may require additional hardware support. Considering the application, rate-based coding with the common neuron model could be easier to deploy. And temporal coding usually requires 2 spikes per neuron on average to encode information [1,2], or single-spike coding with sparsity around 0.6 [3]. So considering the spike number it is possible for rate-based coding to reach the level of temporal coding or be even sparser. In this work, our firing rate during forward is around 7% and with 30 time steps there would be around 2 spikes per neuron on average, and our firing rate during backward is around 3% and with 50 time steps there would be around 1.5 spikes per neuron on average. Though the sparsity may be further improved with training techniques to reach that of IDE, it is acceptable to demonstrate the potential of energy efficiency. We think both rate-based coding with improvement on latency and firing sparsity, and temporal coding are prospective directions, and it should not be “essential to not arrive in a rate-based coding regime”.
>
> [1] Han B. and Roy K. Deep spiking neural network: Energy efficiency through time based coding. ECCV, 2020.
>
> [2] Stockl C. and Maass W. Optimized spiking neurons can classify images with high accuracy through temporal coding with two spikes. Nature Machine Intelligence, 2021.
>
> [3] Zhou S., Li X., Chen Y., et al. Temporal-coded deep spiking neural networks with easy training and robust performance. AAAI, 2021.

---

> > ### Author Response · Authors · 2021-11-17
> > **Response to Review PJRp (2/2)**
> >
> > 2. **Second, we would like to emphasize the problem setting, the contribution of our work and comparison to related works.**
> >
> > The main contribution of this work, which is different from most SNN training methods including Zhou et al. (2021) and Fang et al. (2021), is that we propose a purely spike-based training method with only spike computation of common neuron model (*for both forward inference and backward gradient calculation*) that could achieve high performance with low latency (*for both inference and training*), which provides the potential to conduct energy-efficient training of SNNs on neuromorphic hardware. As described in the Introduce (the second paragraph), most high-performance supervised SNN training methods consider the problem of training SNNs on commonly used computational units (which require complex computation not supported by spike computation to calculate gradients) and deploying trained models for energy-efficient inference. Most existing works including Zhou et al. (2021) and Fang et al. (2021) belong to this category as they require computation that is not spike-based for gradient calculation. Differently, we consider the possibility to purely leverage the same spike computation with common neuron models to implement training for high-performance models, and in a small number of time steps, which has the great potential to perform the training of high-performance SNNs on neuromorphic hardware with energy efficiency. It cannot be realized by existing methods (cf. Table 1). Some previous training methods with spikes such as direct feedback alignment with spikes cannot achieve high performance with low latency. And some other works such as Bellec et al. 2020 consider biologically plausible learning rules that is possible for on-chip learning, which is from another perspective, different from our work that leverages the same spike computation for gradient calculation and training. The comparison shown in Table 1 illustrates the advantage of our method over existing methods. Also as described above, our method could achieve relatively sparse firing with a small number of time steps, which demonstrates the potential of energy efficiency.
> >
> > And compared with IDE (Xiao et al., NeurIPS 2021), IDE only considers the spike computation during forward inference and does not propose the solution to leverage spike computation for backward gradient calculation, so IDE requires training on common computational units; while our work proves that the equilibrium of spike computation can be leveraged to solve implicit differentiation by the design of ternary spiking neuron couples, and we propose to modify the resting membrane potential to make it practical in a relatively small number of time steps. This enables our whole training procedure including both forward inference and backward gradient calculation to be practically based on event-driven spike computation.
> >
> > 3. **About the statement in the abstract.**
> >
> > Sorry for this possibly confusing statement. The statement mainly focuses on recent high-performance supervised SNN training methods in machine learning research, such as converting ANN to SNN or direct SNN training with surrogate derivatives, which require complex computation not supported by spike computation. Bellec et al. 2020 considers the biologically plausible learning rule that is possible for on-chip learning, and is not in the range of this statement. We have revised it into a more precise statement: “However, most high-performance supervised SNN training methods in machine learning research, such as conversion from artificial neural networks or direct training with surrogate gradients, require complex computation not supported by spiking neurons”.
> >
> > 4. **About why negative information is necessary for implicit differentiation calculation.**
> >
> > Because we need to solve a linear equation for implicit differentiation calculation (Eq. (5)): $ \left(J_{g_{\theta}}^\top\vert_{\alpha^*}\right)\beta+\left(\frac{\partial \mathcal{L}(\alpha^*)}{\partial \alpha^*}\right)^\top=0.$, and the elements of the solution $\beta$ could be either positive or negative. If we only have positive information due to positive firing, we cannot leverage the equilibrium of spike rates (which is only positive) to approximate the solution. So we introduce ternary spiking neuron couples to realize ternary outputs in order to solve implicit differentiation. We have added the description in Section 4.1 in the revision.

---

> ### Author Response · Authors · 2021-11-28
> **Please give your further opinions on our paper**
>
> Dear Reviewer PJRp,
>
> We have updated our manuscript and replied to your comments. Would you please check whether our efforts are satisfactory and raise your score? There is only one day left. Many thanks!
>
> Authors

---

### Official Review · Reviewer_dKfz · 2021-11-02

**Correctness:** 3
**Technical Novelty And Significance:** 2
**Empirical Novelty And Significance:** 3
**Recommendation:** 5
**Confidence:** 2

**Main Review:**

This paper proposed a meaningful implementation of the spike-based learning method for SNNs with higher plausibility.
There have been several works on training SNNs with spike-based learning rules based on the back-propagation or the equilibrium propagation, but most of them failed to provide successful experimental results.
It is impressive that the experimental results in this paper showed successful accuracy in several datasets including CIFAR-100.

The writing can be improved for easier understanding.
It is hard to understand basic concepts without reading the reference, so It would be better to explain more details about the background such as the IDE.
Even though I have not fully understood the mathematical derivations in the paper, it was not easy to follow mathematical derivations in the paper without referring to other papers.

========== Comments after rebuttal ==========

I apologize for the late response and thank the authors for adding a detailed explanation of the proposed idea. It was very helpful to deepen my understanding.
I agree that this paper has a clear contribution of proposing a fully spike-based derivation of IDE training methods for SNNs.

However, there remains another concern.
I missed in my initial review that this paper solely used the IF neuron model. Can the proposed method also be applied to other spiking neuron models?

Even though the IF neuron model is 'common' in many SNN literature, I strongly believe that the study on SNN is meaningful only when temporal information carried by spikes is utilized in information processing. Averaging out spikes with rate-coding with IF neuron is functionally identical to quantized non-spiking neural networks, in which the activation function has quantized output. Multiplication with linearly-quantized values (e.g. low-bit fixed-point numbers) is already cost-effective and can also be implemented only by accumulation. I don't think there is a justifiable reason to use SNNs instead of this counterpart which can simply use conventional hardware. Major benefits of neuromorphic hardware such as event-drivenness can also be replaced by the zero-skipping feature that is common in NN hardware. Even though the proposed training method is fully spike-based, with the IF neuron model, it seems not to use any temporal information of spikes.

I would like to hear the authors' opinions on this. Thank you.

**Summary Of The Paper:**

In this paper, the authors proposed a method to train Spiking Neural Networks (SNN) with spike-based implicit differentiation on the equilibrium state.
Main idea is to use a spike-triggered event instead of average firing rate to approximate implicit differentiation of Feedback Spiking Neural Networks (FSNN).
To enable such idea and to further reduce the approximation errors, the authors proposed several techniques such as adopting ternary spiking neuron couples and shifting resting potential.
The experimental results showed that the proposed method can achieve high accuracy in several tasks such as MNIST, CIFAR-10, and CIFAR-100 with fewer time steps for training compared to existing methods.

**Summary Of The Review:**

The proposed method effectively handles several obstacles while adopting the existing IDE method with spike-based implementation and provided convincing experimental results.
However, I doubt that the theoretical improvements of this paper is significant enough especially with my poor understanding on the mathematical derivations in the paper.

---

> ### Author Response · Authors · 2021-11-17
> **Response to Reviewer dKfz (1/2)**
>
> Thank you very much for your valuable comments. We try our best to address your concerns as follows.
>
> 1. **More details about the background such as the IDE.**
>
> Thank you very much for your suggestion. Due to the space limit, we only briefly introduced the background in Section 3. We have added more descriptions about the background in Section 3 and in Appendix A for easier understanding in the revision. We quote them here:
>
> *Due to the complex spiking neuron model which is discontinuous, directly supervised training of SNNs is a hard problem, since the explicit computation is non-differentiable and therefore backpropagation along the forward computational graph could be problematic. The IDE training method (Xiao et al., 2021) considers another approach to calculating gradients that does not rely on the exact reverse of the forward computation, which avoids the problem of non-differentiability as well as large memory costs by BPTT-like methods with surrogate gradients. Specifically, the IDE training method first derives that the (weighted) average firing rate of FSNN computation with common neuron models would gradually evolve to an equilibrium state along time, which follows a fixed-point equation. Then by viewing the forward computation of FSNN as a black-box solver for this equation, and applying implicit differentiation on the equation, gradients can be calculated only based on this equation and the (weighted) average firing rate during forward computation rather than the exact forward procedure. Therefore, the forward and backward procedures are decoupled and the non-differentiability is avoided.*
>
> *As introduced in Section 3.2, the IDE method defines the average firing rates of spikes during forward computation, i.e. $\alpha[t]$. Then IDE could derive an equivalent update equation for $\alpha[t]$, based on the integrated update equations of membrane potentials Eq. (2,3). With the equivalent equation for $\alpha[t]$, IDE proves that under certain conditions, $\alpha[t]$ converges to an equilibrium state following a fixed-point equation. With the assumption that $\alpha[T]$ after simulation of $T$ time steps roughly follows the fixed-point equation of the equilibrium state, gradients of the loss function for parameters can be calculated by implicit differentiation, as introduced in Section 3.3. So the training pipeline for the IDE method can be summarized as: first simulate FSNN computation for $T$ time steps to obtain the rate of spikes $\alpha[T]$, then solve the implicit differentiation by root-finding methods for gradient calculation based on $\alpha[T]$ and the derived fixed-point equation of equilibrium states, and finally apply gradient-based optimizers to update parameters.*
>
> 2. **About the mathematical derivation.**
>
> For easier understanding, we add a brief outline for the mathematical derivation in Section 4.2 in this revision: we first derive the update equation of membrane potentials based on the basic SNN computation (Eq. (7) and (10)), then we derive the equivalent equation of the average firing rate of spikes based on the above equation and the definition (Eq. (8)), following which we make a division of the term $u$ in the equation to derive another equivalent equation with eliminating perturbation based on our designed ternary neuron couples (Eq. (9) and (11)), and finally based on it, we could prove that under certain conditions, the average firing rate of spikes would converge to an equilibrium, which is the solution of the equation for implicit differentiation calculation (Theorems 1 and 2).
>
> As for the derivation of approximation error with finite time steps in Section 4.3, we describe the basic outline as follows for easier understanding. We decompose the error into several components, and the main error we consider to reduce is from the perturbation term $u$ in the derived equivalent equation of the average firing rate of spikes (Eq. (8)). We mainly analyze this term from the statistical perspective, and find that the mean and variance of the random error is related to the resting potential of SNNs (given threshold). Then based on the analysis, we could propose to modify the resting potential to reduce the approximation error.

---

> > ### Author Response · Authors · 2021-11-17
> > **Response to Reviewer dKfz (2/2)**
> >
> > 3. **About the theoretical improvement.**
> >
> > As for the theoretical improvement compared with IDE (Xiao et al., NeurIPS 2021), we prove that implicit differentiation can be solved by common spiking neuron models based on our ternary neuron couples, so that it is possible for spike-based training on neuromorphic hardware, and we propose how to reduce the approximation error with finite time steps based on the theoretical analysis on it, so that spike-based training could be practical in a small number of time steps for energy efficiency. IDE requires general root-finding methods with complex computation to solve implicit differentiation, which is not spike-based (cf. Table 1). And it remains unclear if implicit differentiation could be solved by common spiking neuron models with only positive firing. Our work design ternary spiking neuron couples and prove that the equilibrium of spike computation can be leveraged to solve implicit differentiation based on the design. And to make it practical in a relatively small number of time steps, we analyze the approximation error from the statistical perspective and propose to modify the resting membrane potential to reduce it. This enables our whole training procedure to be practically based on event-driven spike computation, which is our unique contribution.

---

> ### Author Response · Authors · 2021-11-28
> **Please give your further opinions on our paper**
>
> Dear Reviewer dKfz,
>
> We have updated our manuscript and replied to your comments. Would you please check whether our efforts are satisfactory and raise your score? There is only one day left. Many thanks!
>
> Authors

---

> ### Author Response · Authors · 2021-11-30
> **Further response to Reviewer dKfz (1/2)**
>
> Thank you very much for your valuable comments. We try our best to address your concern as follows.
>
> 1. Our method can be applied to LIF model with weighted average firing rate coding which leverages the temporal information as well. The original IDE method (Xiao et al., 2021) has derived the equilibrium state of forward computation with both IF model (w.r.t average firing rate) and LIF model (w.r.t weighted average firing rate), where for LIF model, spikes at different time steps have different weights according to the response kernel of LIF model. The weighted average firing rate is defined as $\mathbf{\hat{a}}[t]=\frac{\sum_{\tau=1}^t \lambda^{t-\tau}\mathbf{s}[\tau]}{\sum_{\tau=1}^t \lambda^{t-\tau}}$, where $\lambda$ is the leaky term in the discrete computational form. Since we extend the IDE method mainly on the spike-based gradient calculation, we can leverage the LIF model during forward calculation to enable utilizing temporal information of the model. This can lead to the basic method with LIF model for forward while IF model for backward.
>
> Additionally, we may similarly leverage LIF model for backward and derive the equilibrium state (w.r.t weighted average firing rate) for solving implicit differentiation. In this setting, similar to the conclusion of equilibrium states in the IDE method, there might be random error for approximating the equilibrium state. Due to this reason, originally, we first focus on IF model and also leverage IF model during forward for consistency. Actually, our method can be applied to LIF model (forward LIF backward IF, or forward LIF backward LIF), as shown in the results presented below.
>
> We supplement some preliminary results with LIF model below. The experiments are on MNIST with the same feedback structure in Table 3. All hyperparameters are the same as in the paper and we set the leaky term as $\lambda=0.95$.
>
> | Forward model | Backward model | $T_F$ | $T_B$ | Accuracy
> | :----: | :----: | :----: | :----: | :----: |
> | IF | IF | 30 | 100 | 99.34%$\pm$0.02% (99.37%) |
> | LIF | IF | 30 | 100 | 99.32%$\pm$0.04% (99.37%) |
> | LIF | LIF | 30 | 100 | 99.34%$\pm$0.05% (99.39%) |
>
> It shows that our method is still effective for LIF model. Due to the limited time, only one run of results on MNIST are available now (**update**: we update results with 3 runs of experiments). We will add the discussion, derivation, details, and more results in the follow-up revision.
>
> Therefore, our method can be spike-based with LIF model and utilization of temporal information as well, and is not limited to IF model that, as your comment, does not fully take advantage of SNN computation. It is also interesting to further consider how to utilize more temporal information of SNN in future work, which is an open question for SNN as well.

---

> > ### Author Response · Authors · 2021-11-30
> > **Further response to Reviewer dKfz (2/2)**
> >
> > 2. Additionally, SNN with IF neuron may be applicable to more **flexible network structures** compared with quantized non-spiking neural networks, such as including **feedback connections**. Please note that “averaging out spikes with rate-coding with IF neuron is functionally identical to quantized non-spiking neural networks” is correct only when we consider the feedforward structure, and if we consider feedback structures and equilibriums, they are not functionally identical. Especially, since the computation of SNN naturally involves multiple time steps, introducing feedback connections hardly requires additional costs. But for quantized non-spiking neural networks, we will have to introduce the new concept of time steps to realize feedback connections, which requires much more additional costs, and is not exactly functionally identical to feedback spiking neural networks. As we have mentioned in the third paragraph in Introduction, most recent SNN works simply imitate feedforward structures from ANNs and ignore the feedback connections that are potentially more suitable for SNNs. In fact, considering the feedback network structure may, to some extent, be viewed as leveraging temporal information of SNNs, which is not from the perspective of information coding but from the perspective of network structures.
> >
> > In this work, our method basically focuses on the feedback network structures, and can also be degraded to feedforward ones, as introduced in Introduction. Therefore, even considering SNN with IF neuron, our model cannot be simply replaced by quantized non-spiking neural networks, which may require more costs to introduce feedback connections and equilibriums. Could it be a justifiable reason to use feedback SNNs even with IF neuron?

---

> > > ### Comment · Reviewer_dKfz · 2021-12-01
> > > **Response to the comment**
> > >
> > > 1. I thank the authors for conducting experiments for my concern in a very short time. It is impressive that the proposed method works even with the leaky nature of the spiking neurons. However, as the authors mentioned as "random error for approximating the equilibrium state", when using averaged-out spike rate (especially at the equilibrium), the temporal information of the spikes are only noise that may be useful only for regularization. In the network structure with deeper and larger network structures, this noise can be more problematic for training. I understand that the proposed method is in the early stage of study and do not demand the results with larger networks, but I wonder if the authors have any idea that the temporal information of the spikes (e.g. leakage or specific spike timings) is meaningfully used with the proposed method or just with the training regime that uses the converged spike rate at the equilibrium.
> > >
> > > 2. I clearly overlooked the importance of the feedback connections in the temporal information in SNNs. I definitely understand that when the network structure with feedforward/feedback connections is complicatedly connected across the layers SNNs can be a more efficient implementation compared to the quantized NNs. However, I am not certain about the need for feedback connections when using the spike rate at the equilibrium. It seems like the only reason for using the feedback connections in such a setting is to flow information from the last layer for training. To leverage temporal information of the feedback connections, the transient fluctuations of membrane potentials or the spike rates should be used for any purpose. Even if the proposed method can be seamlessly applied to the much complicated network structure, I question that such complicated network structure can utilize the complex temporal information when only the converged spike rate is used for output.
> > >
> > > I would like to thank the authors for constructive discussion.

---

> > > > ### Author Response · Authors · 2021-12-01
> > > > **Further response to Reviewer dKfz**
> > > >
> > > > Thank you for your valuable comments.
> > > >
> > > > 1. For our LIF model, the temporal information is not only noise, but has the effect on the spike representation. For LIF model, we consider the **weighted** average firing rate as the spike representation rather than averaged-out spike rate. The weighted average firing rate is defined as $\mathbf{\hat{a}}[t]=\frac{\sum_{\tau=1}^t \lambda^{t-\tau}\mathbf{s}[\tau]}{\sum_{\tau=1}^t \lambda^{t-\tau}}$ (it is the discrete form and the continuous form would be $\mathbf{\hat{a}}(t)=\frac{\int_0^t \kappa(t-\tau)\mathbf{s}(\tau)\mathrm{d}\tau}{\int_0^t \kappa(t-\tau)\mathrm{d}\tau}$), so spikes at different time steps have different weights to the spike representation according to the response kernel with leakage of LIF model. This can be realized by a leaky readout non-spiking neuron with accumulated membrane potentials, as mentioned in Xiao et al. 2021. Therefore, the temporal information can be leveraged to enlarge the representation space, because for simply averaged-out spike rate, spikes are equivalent at any time step for a representation 1/TimeStep, while for weighted average firing rate, spikes at different time steps can be used to represent different values according to the leakage. As for the noise, it is in the context of the limit, and since we want a low latency, there already exists noise caused by small finite time steps for both IF and LIF model, and we observe that the additional random noise of LIF model do not harm the training. Instead, the weighted spike representation may enhance LIF model with low latency due to the temporal information. We will consider the generalization to larger models and larger datasets in future work. And as for the specific spiking timings, we have considered if it can be leveraged to introduce more temporal information. It seems that leveraging the specific time ideally requires the continuous-time setting, or at least with small discrete time intervals, which may need a large number of time steps and would be a large latency with existing discrete simulation of SNNs. With the consideration of low latency in discrete setting, the specific spiking time may have the similar effect to average rate coding, as the time interval is with the precision 1/TimeStep as well. So we mainly consider using the leakage temporal information for our spike representation. It would be interesting to consider more possible coding schemes in future work.
> > > >
> > > > 2. Our feedback connections are to introduce dynamics and equilibriums for network computation. Feedforward structures are direct functional mapping, while feedback systems involve dynamics and equilibriums, which may lead to rich representations. And as introduced in the third paragraph, feedback connections may enable neural networks to be shallower and more brain-like [1]. Our consideration of feedback connection is from the perspective of network structures, and temporal information is leveraged as in the above response. Such flexible network structures suits SNNs more compared with quantized non-spiking neural networks. It would be interesting future work to further consider, e.g. temporal delay of feedback connections, which may result in different dynamics and equilibriums to leverage more temporal information.
> > > >
> > > > Hope it could address your concerns.
> > > >
> > > > [1] Jonas Kubilius, Martin Schrimpf, Kohitij Kar, et al. Brain-like object recognition with high-performing shallow recurrent anns. NeurIPS, 2019.

---

### Official Review · Reviewer_FWiT · 2021-11-02

**Correctness:** 3
**Technical Novelty And Significance:** 2
**Empirical Novelty And Significance:** 2
**Recommendation:** 6
**Confidence:** 4

**Main Review:**

The authors attempt to support their conclusion with mathematical evidence, which improves the reliability of the quantitative part of the results. Yet every coin has two sides. At the same time, the math details cannot tell the upper bound and the potential of the proposed method, which is my main concern for the series of approaches.

In this paper, the authors extend the work Xiao et. al., 2021 NeurIPS to elaborate on the ideas of training SNN based on the equilibrium and address some of the shortcomings. Overall, this paper is clearly written and easy to follow. I list my comments and concerns below and am more than happy to discuss with the authors about these issues.

1. I think the idea of approaching SNN training via equilibrium is interesting, but it requires more reasoning about why it has the potential to go beyond existing methods and what is its unique methodological contribution given Xiao et. al., 2021 NeurIPS.

2. If I understand it correctly, the equilibrium basically says that an integrally convergent input would result in an integrally convergent average firing rate. Since the assumption satisfies Lipschitz's condition with L < 1, I feel like the prove could be simplified with some known ODE theorems on perturbation.

3. The derivative on the equilibrium gives the dependence between the convergent $a^*$ and $x^*$. I think it is more related to backpropagation rather than the  Hebbian learning rule. The analog here is not evident.

4. Since the whole framework is built on the equilibrium, it implicitly requires the dynamics to be stable. Thus, it seems that the latency $T_F$ cannot be extremely shortened to an efficient case where the dynamics are not stable across time. As the authors also pointed out, this paradigm is not compatible with setups that may amplify the norm and maps, e.g. the BN-layers in training. This shortcoming from assumption may limit the practical utility of the proposed methods. In practice, the comparison in Table 3 is not up-to-date. For example, Ref [1] and [2-3] significantly shorten the conversion and training latency and work well for big datasets. The experiments in the current work should at least extend to CIFAR-100 and ImageNet. It would be also great if the authors can additionally add the results on the spiking dataset like CIFAR-gesture and CIFAR10-DVS as well.

5. in Figure 3, it seems that the accuracy has a jump around epoch 50. Can the author explain the reason?
------------------------------------------------------------------------------------------------------------------------------------------------------
[1] Yuhang Li, Shikuang Deng, Xin Dong, Ruihao Gong, Shi Gu. A Free Lunch From ANN: Towards Efficient, Accurate Spiking Neural Networks Calibration. *Proceedings of the 38th International Conference on Machine Learning, PMLR 139:6316-6325, 2021.*

[2] Fang, Wei, Zhaofei Yu, Yanqi Chen, Tiejun Huang, Timothée Masquelier, and Yonghong Tian. Deep residual learning in spiking neural networks. *Conference on Neural Information Processing Systems (2021). *

[3] Wu, Jibin, Yansong Chua, Malu Zhang, Guoqi Li, Haizhou Li, and Kay Chen Tan. A tandem learning rule for effective training and rapid inference of deep spiking neural networks. *IEEE Transactions on Neural Networks and Learning Systems (2021).*

------------------------------------------------------------------------------------------------------------------------------------------------------
My comments after the rebuttal:

The authors made a very attentive rebuttal and emphasized their contribution as a purely spike-based training approach. Yet my two major concerns still remain here.
1. I cannot fully track every step in the updating formula to make sure that it is purely spiked-based. For example, does the calculation of (10) and (11) and the associated gradient calculation on page 7 still involve multiplication? If the general multiplication is somewhat supported by neuromorphic hardware, the other methods mentioned in the comparison would also work as well?
2. If we accept the claim it is purely spike-based, we still have the question of whether the potential energy efficiency is worthy. For example, the performance on DVS-CIFAR-10 is ~15\% less than the mentioned benchmarks on page 22. As the authors mentioned, the energy cost is related to both the latency and spiking rate. Thus it would be necessary to at least provide a comparison of the estimated cost with the relevant approaches and better to provide an implementation of at least the proposed method on proper hardware to support the purely-spike-based assertion. Also, it would be necessary to demonstrate that the proposed method can achieve at least comparable accuracy with Fang et. al., ICCV 2021 to support that the cost of changing to spike-based training is acceptable.

Based on these two points, I tend to keep my rating at 5 but am open to marginally accepting this paper if the other reviewers strongly think it should be accepted.
------------------------------------------------------------------------------------------------------------------------------------------------------
Latest update:

I encouragingly increased my score to 6 given the rebuttal effort but my judgment remains on the borderline as the major concerns are not fully addressed with sufficient numeric proof.



**Summary Of The Paper:**

In this paper, the authors extend the work Xiao et. al., 2021 NeurIPS to elaborate on the ideas of training SNN based on the equilibrium and address some of the shortcomings.


**Summary Of The Review:**

Overall, this paper is clearly written and easy to follow, but the methods may not apply to the more complex scene with less latency. My concern is that this limitation is from its setup.

---

> ### Author Response · Authors · 2021-11-17
> **Response to Reviewer FWiT (1/2)**
>
> Thank you very much for your valuable comments. We try our best to address your concerns as follows.
>
> 1. **About the contribution and the potential.**
>
> First, we would like to clarify and emphasize the problem setting and the contribution of this work. The main contribution of this work, which is different from most SNN training methods including IDE (Xiao et al., NeurIPS 2021), is that we make attempts to perform the *whole training* of SNNs purely based on *spike computation*, *including both forward inference and backward gradient calculation*, which provides the potential to conduct energy-efficient training of SNNs on neuromorphic hardware. As described in the Introduction (the second paragraph), most SNN training methods consider the problem of training SNNs on commonly used computational units (which require complex computation not supported by spike computation to calculate gradients) and deploying trained models for energy-efficient inference. Differently, we consider the possibility to carry out the whole training procedure based on spike computation, which has the great potential to directly perform the training of SNNs on neuromorphic hardware with energy efficiency. So the basic focus of this work is how to purely leverage spike computation with common neuron models to implement training for high-performance models in a small number of time steps. It cannot be realized by existing methods (the comparison shown in Table 1 illustrates the advantage of our method over existing methods.).
>
> As for the uniqueness compared with IDE (Xiao et al., NeurIPS 2021), there are methodological contributions to solving implicit differentiation by common spiking neuron models and reducing approximation error. The basic thought of IDE is to identify the underlying equilibrium states of FSNN computation during forward inference, so that gradients can be calculated by implicit differentiation. However, IDE requires general root-finding methods with complex computation to solve implicit differentiation, which is *not* spike-based (cf. Table 1). While the IDE paper briefly conjectured the equilibrium computation for implicit differentiation, it remains unclear if it could be solved by common spiking neuron models with only *positive* firing. Our work extends the thought of equilibrium of spike computation to solving implicit differentiation by the design of ternary spiking neuron couples, and we propose to modify the resting membrane potential to make it practical in a relatively small number of time steps. This enables our whole training procedure to be based on event-driven spike computation, which is our unique contribution.
>
> 2. **“Feel like the prove could be simplified with some known ODE theorems on perturbation”.**
>
> ODE theorems may not be directly applied to the ODE for membrane potentials with discontinuous spike generation (i.e. transitions). Actually, we do not directly deal with this ODE, but consider the integrated update equations for firing rates, which is more similar to the iterative fixed-point update scheme. So our proof is similar to that of the contraction mapping theorem (with perturbation), which is not very complicated but we present the full details.
>
> 3. **“It is more related to backpropagation rather than the Hebbian learning rule. The analog here is not evident.”**
>
> What we would like to suggest is that the weight update is proportional to the two-stage firing rates of the two connected neurons respectively, which is local after two stages and might share some kind of thought of original locally updated learning rules and be possible on neuromorphic hardware. As it may read confusing, we have modified the description without mentioning the Hebbian learning rule in the revision.
>
> 4. We respond to each question as follows.
>
> 4.1 **About the practical utility.** As explained in the response to Question 1, the main focus and contribution of this work is to purely leverage spike computation with common neuron models to implement training for high-performance models, which provides the potential to conduct energy-efficient training of SNNs on neuromorphic hardware. It is due to the consideration of possible neuromorphic operations that some setups like BN which requires batch statistics information may not be applicable. We may seek alternative normalization methods that are friendly for neuromorphic implementation. Besides, it has another potential to finetune deployed models directly on neuromorphic hardware (which may be pre-trained with BN on common computational units previously) for new data or new tasks, e.g. for continual learning. So there could be practical utility, and we mainly demonstrate the effectiveness of the training method in this work while those aspects could be prospects.

---

> > ### Author Response · Authors · 2021-11-17
> > **Response to Reviewer FWiT (2/2)**
> >
> > 4.2 **About the latency.** It is true that equilibrium requires some latency. In practice, $T_F=30$ could work well, which is actually a relatively small number of time steps. We do not aim at “extremely low” latency, because of the following reason: the real simulation time is related to not only the reported latency but also network scale and depth, and the energy consumption is additionally related to firing rate. Currently, the latency reported by most papers is the time step at each layer, which does not take network depth into account. Actually, information should be propagated successively along with layers, so the total simulation time steps would be the reported time step plus the layer number (if it takes 1 time step to propagate information along with layers, the real time may also be related to computational costs). Therefore, seemingly extremely low latency with a very deep network, e.g. 5 time steps for 100 layers, may take more simulation time than moderate latency with a shallow network, e.g. 30 time steps for 5 layers. And the consideration of feedback network structures and equilibriums could aim at relatively shallow structures, which can be viewed as turning depth into time steps. Besides, the energy consumption of SNNs should be proportional to the spike number, the product of latency and average firing rate. So it would be a balance between accuracy, network scale, latency, as well as firing rate, and the latency is not the only criterion.
> >
> > 4.3 **About experiments.** Our experiments are mainly to demonstrate the potential of energy-efficient training considering the firing rate and time steps, as well as the effectiveness of our method. We do not aim at outperforming the recent state-of-the-art results, but just show that a comparable performance could be achieved even with our constraints of purely spike-based training in a relatively small number of time steps, verifying the effectiveness. So we mainly compare results with similar network structures as ours. The results of **CIFAR-100** are already included in the paper (in Section E.3 in Appendix (in the revised version it will be Appendix F.3), we have mentioned in the main text, line 11 in the last paragraph of Section 5, that the results are in Appendix due to the space limit). We also present the results here. The performance of our model is competitive for networks without BN and our model is with fewer neurons and parameters and a small number of time steps.
> >
> > | Method | Network structure | BN | $T_F$ | $T_B$ | Mean$\pm$Std (Best) | Neurons | Params
> > | :----: | :----: | :----: | :----: | :----: | :----: | :----: | :----: |
> > | BP (Thiele et al., 2019a) | CIFARNet | no | Unkown | / | (64.69%) | 726K | 45M |
> > | IDE (Xiao et al., 2021) | CIFARNet-F | yes | 30 | / | 71.56%$\pm$0.31% (72.10%) | 232K | 14.8M |
> > | SpikeGrad (Thiele et al., 2019a) | CIFARNet | no | Unknown | Unknown | (64.40%) | 726K | 45M |
> > | SPIDE (ours) | CIFARNet-F | no | 30 | 100 | 63.57%$\pm$0.30% (63.91%) | 232K | 14.8M |
> > | SPIDE (ours) | CIFARNet-F | no | 30 | 250 | 64.00%$\pm$0.11% (64.07%) | 232K | 14.8M |
> >
> > Not many direct SNN training methods conduct experiments on ImageNet due to the large training costs, and the benchmark datasets we choose are commonly used in the existing literature. We think that the current results are adequate to support the verification of the potential energy efficiency and effectiveness of our method.
> >
> > Additionally, we supplement the result on the spiking dataset **CIFAR10-DVS** at the request. The details and discussion are in Appendix F.4 in the revised version, and the result is below:
> >
> > | Method | Model | $T_F$ | $T_B$ | Accuracy
> > | :----: | :----: | :----: | :----: | :----: |
> > | Gabor-SNN (Sironi et al., 2018) | Gabor-SNN | / | / | 24.5%|
> > | HATS (Sironi et al., 2018) | HATS | / | / | 52.4%|
> > | STBP (Wu et al., 2019) | Spiking CNN (LIF, w/o NeuNorm) | 40 | / | 58.1%|
> > | STBP (Wu et al., 2019) | Spiking CNN (LIF, w/ NeuNorm) | 40 | / | 60.5%|
> > | Tandem Learning (Wu et al., 2021) | Spiking CNN (IF) | 20 | / | 58.65%|
> > | Spike-based BP (Fang et al., 2021) | Spiking CNN (PLIF, w/ BN) | 20 | / | 74.8%|
> > | SPIDE (ours) | Spiking CNN (IF) | 30 | 250 | 60.7%|
> >
> > Our result is competitive among results of common SNN models, demonstrating the effective of our method. Fang et al. (2021) leverages many techniques, such as learnable membrane time constant, batch normalization, max pooling and so on to achieve better performance. We do not aim at outperforming the state-of-the-art results, but demonstrate that a competitive performance could be achieved even with our constraints of purely spike-based training.
> >
> > 5. **“Why the accuracy has a jump around epoch 50.”**
> >
> > It’s because we decay the learning rate at epochs 50 and 75. We use the “step” learning rate schedule, which is a common practice for NN training, and the accuracy jump is a common phenomenon.

---

> ### Author Response · Authors · 2021-11-28
> **Please give your further opinions on our paper**
>
> Dear Reviewer FWiT,
>
> We have updated our manuscript and replied to your comments. Would you please check whether our efforts are satisfactory and raise your score? There is only one day left. Many thanks!
>
> Authors

---

> ### Author Response · Authors · 2021-11-30
> **Further response to Reviewer FWiT**
>
> Thank you for your further comments. We try our best to address your concerns as follows.
>
> **1. About whether purely spike-based.**
>
> Our proposed training method is based on spike computation and does not require multiplication in Eq. (10). Please note that the notation for matrix-vector multiplication in Eq. (10) just represents the spike communication between spiking neurons, and s[t] are spikes so the ‘multiplication’ can be realized by accumulation triggered by spikes, which is the same as the common forward SNN computation, and M is a binary mask on weights as defined. And Eq. (11) is a derived equivalent formulation for ‘average firing rates’ based on the calculation of Eq. (10), which is just used to derive the equilibrium states in Theorems and is not calculated in practice. As for the gradient calculation on page 7, as we have discussed in the first paragraph on page 7, “during the calculation of gradients based on two-stage average firing rates, since we will have the firing rate of the first stage before the second stage, this calculation can also be carried out by event-based calculation triggered by the spikes in the second stage."
> Other SNN training methods do not consider spike-based computation during gradient calculation, and involve matrix multiplication (and perhaps calculation of surrogate derivative for the step function), which is not supported by spike-based neuromorphic hardware.
>
> **2. About the energy cost comparison.**
>
> We provide some theoretical estimation of energy costs.
>
> As demonstrated in Section 5, the average firing rate of our method during backward calculation is around 3%, and we can still obtain competitive results with $T_B=50$. Then each neuron will generate 1.5 spikes on average in total. Each connection between the neuron and other neurons will perform 1.5 AC operations (accumulation) on average.
>
> While for other training methods such as the representative STBP method, if we simply do not consider the cost for calculation of surrogate derivative for the step function and just consider the matrix calculation during backward, then each connection of neurons will perform one MAC operation (multiply and accumulate) at every forward simulation time step (because STBP needs backpropagation along the computational graph unfolded along time). So each connection will perform $1\times T_F$ MAC operations.
>
> According to the energy consumption of the 45nm CMOS process, the energy for 32bit FP MAC operation is 4.6 pJ, and the energy for 32bit FP AC operation is 0.9. So theoretically the energy consumption of our method during backward will at least be **$\frac{4.6\times T_F}{0.9\times 1.5}=3.4\times T_F$ times less** than BPTT-like methods (if they choose $T_F=12$, then it is **$40\times$ reduction**). So we think the potential energy efficiency is promising.
>
> As for the implementation on proper hardware, sorry that the limited time does not allow us to do so. This work mainly focuses on the algorithm, and the implementation on hardware could be future work.
>
> **3. About the performance on DVS-CIFAR10.**
>
> Please note that the ‘benchmark’ of Fang et al. is the state-of-the-art result just released this year (ICCV 2021), and in their comparison, they outperform the previous SOTA result by 14.3%. They leverage many techniques to improve the results. We do not carefully tune the hyperparameters or study and adopt their techniques, but just demonstrate that similar performance could be achieved compared with previous typical results such as STBP (Wu et al., 2019) and tandem learning (Wu et al., 2021). It could be future work to further improve the results with more techniques.

---

> > ### Comment · Reviewer_FWiT · 2021-11-30
> > **A few further thoughts**
> >
> > 1. In terms of the estimation, I think it should be flop-based and it is a bit strange that the proposed method's cost is unrelated to the $T_F$. So I feel like a more realistic estimation should be provided here. For example, you can refer to [1] for the estimation and [2] for a plan of detailed demonstration of the computing efficiency. I proposed this concern because the purely-spiked training is the major selling point of this paper thus demands sufficient verification at least heuristically to demonstrate its superiority over other SNN training approaches. Also, I do not quite understand “during the calculation of gradients based on two-stage average firing rates, since we will have the firing rate of the first stage before the second stage, this calculation can also be carried out by event-based calculation triggered by the spikes in the second stage." Shouldn't we combine the cost in these two stages together?
> >
> > 2. In terms of the performance, I did mean you should cite or beat this SOTA although it was actually posted online last year. My intuitive thought here is the work reviewed around the period of time should achieve a similar level of performance and it is the author's obligation to demonstrate whether the proposed one can be coherently integrated with the existing ones so that the progression is addable or monotonously increasing for the field.
> >
> > -----------------------------------------------------------------------------
> >
> > I would love to encouragingly increase my score to 6 given the authors' efforts in the rebuttal but strongly suggest that the authors should technically address these two points in the next version no matter it is camera-ready or re-submission as it is critically important to support the claimed contribution of the current work.
> >
> > [1] Nitin Rathi and Kaushik Roy. Diet-snn: Direct input encoding with leakage and threshold optimization in deep spiking neural networks. arXiv preprint arXiv:2008.03658, 2020.
> > [2] Wu, Jibin, Yansong Chua, Malu Zhang, Guoqi Li, Haizhou Li, and Kay Chen Tan. A tandem learning rule for effective training and rapid inference of deep spiking neural networks. IEEE Transactions on Neural Networks and Learning Systems (2021).

---

> > > ### Author Response · Authors · 2021-11-30
> > > **Thank you for valuable suggestions and further response**
> > >
> > > Thank you very much for your valuable comments and suggestions.
> > >
> > > 1. We would like to clarify our estimation of energy costs. In the response, we present the estimated energy costs **during backward gradient calculation**, since we do not modify the forward SNN computation and therefore the energy costs for the forward calculation should be the same among different methods if deployed with spike-based computational hardware. So we mainly present how our method could theoretically save the energy of the backward stage to demonstrate the potential of spike-based training. And our backward stage is only related to $T_B$ rather than $T_F$, and the influence of $T_B$ is included in the calculation of spike numbers. The backward stage of STBP is related to $T_F$ because their forward computational graph is unfolded by $T_F$ time steps, so their BPTT should be backpropagated $T_F$ times. In the response, we calculate the number of spikes per neuron on average by multiplying the firing rate by $T_B$, which is the same as SpikeRate in [1], i.e. the involved accumulation operations. And combined with the energy estimation of operations, a clearer calculation is: $\frac{E_{ours}}{E_{STBP}}=\frac{OPNum \times E_{AC}}{OPNum \times E_{MAC}}=\frac{FiringRate \times T_B \times E_{AC}}{T_{F_{STBP}} \times E_{MAC}}=\frac{0.03 \times 50 \times 0.9}{12 \times 4.6}=\frac{1}{40}$. As for the sentence “during the calculation …”, we mean that this gradient calculation can be carried out by spike-based computation as well. The two stages correspond to the forward and backward stages, and as explained above, the forward stage is the same for SNN models (which achieves energy-efficient inference compared with ANNs, and that’s what the energy analysis in [1] focuses on), and we mainly focus on the energy analysis for backward stage which is our uniqueness. Additionally, if we consider the setting that the forward computation of STBP also has to be simulated by, e.g. GPU, to be compatible with backward in order to realize training, while ours may be deployed with neuromorphic hardware, then the forward stage during training can also enjoy the energy efficiency as $\frac{E_{ours}}{E_{STBP}}=\frac{OPNum \times E_{AC}}{OPNum \times E_{MAC}}=\frac{FiringRate \times T_F \times E_{AC}}{T_{F_{STBP}} \times E_{MAC}}=\frac{0.07 \times 30 \times 0.9}{12 \times 4.6}=\frac{1}{29}$ (based on the firing rate around 7% as in Figure 2 and $T_F=30$). We will add a detailed discussion in the next version. Thank you for your suggestion.
> > >
> > > 2. As for the performance, we will try our best to study and adopt the advanced techniques. Thank you for your suggestion. We still want to have a little discussion of our focus. The basic focus of this work is not outperforming SOTA results, but exploring the new possibility to perform spike-based training with energy efficiency and verifying the effectiveness with common settings. There usually exists a balance for new exploration. For example, currently directly trained SNNs can still hardly reach the performance of SOTA ANNs, or SOTA conversion from ANNs, especially on large-scale datasets, and not all abundant advanced techniques in ANNs are adopted in SNN papers (e.g. advanced data augmentation or regularization techniques). Despite this, direct training of SNNs explores the possibility to perform low latency inference for energy efficiency, and we can accept them without strict requirement to outperform SOTA ANNs with numerous techniques which may need further tuning for adapting to SNNs. Anyway, we will try our best to study the advanced techniques and further improve the performance. Thank you for your suggestion.

---

### Author Response · Authors · 2021-11-17
**A summary of paper updates**

We thank all reviewers for their valuable comments. We have uploaded the updated version of our paper based on the reviews. Revisions are marked as blue in the text. The updates are summarized as follows:

1. In response to Reviewer PJRp, we revise the description regarding SNN training methods in the abstract into a more precise statement.

2. In response to Reviewer FWiT, we supplement the results on CIFAR10-DVS in Appendix F.4, and add several descriptions in the main text (Section 5).

3. In response to Reviewer dKfz, we add more descriptions about the background in Section 3 and in Appendix A for easier understanding. We also add a brief outline for the mathematical derivation for easier understanding in Section 4.2.

4. In response to Reviewer PjRp, we add the explanation about why we need negative information for implicit differentiation calculation in Section 4.1.

5. In response to Reviewer FWiT, we revise the description about the analog to Hebbian learning rule in Section 4.2.

6. We add more descriptions about the contribution of this work compared with IDE in Appendix A. We also modify several descriptions in the Introduction to emphasize the problem setting of this work.

---

### Decision · Program_Chairs · 2022-01-20

**Decision:**

Reject

**Comment:**

The paper introduces a purely spike based method for training spiking neural networks with recurrence, by extending the recently published "implicit differentiation on the equilibration state (IDE)" technique. As a purely spike based method for both the forward pass and the gradient computation, the proposed technique potentially represents an important advance.

Based on the original submission, the reviewers had difficulties understanding the paper's contributions and verify its claims. I commend the authors for engaging with the reviewers by answering their questions and updating the paper to better explain the contributions. However, even after considerable back and forth, the most positive reviewer still expressed major concerns [[1](https://openreview.net/forum?id=VQyHD2R3Aq&noteId=Ubrdawfds5)], and the other reviewers appeared unmoved, based on their scores.

The reviewers' principal concerns were a little hard to distill. It is possible that their initial difficulties with understanding and validating the paper's contributions made it hard to fully appreciate the paper. One reviewer is unable to verify that the algorithm is purely spike based, and that the energy costs are appropriately calculated. They are also unsure if the method will scale to more complex settings [[1](https://openreview.net/forum?id=VQyHD2R3Aq&noteId=Ubrdawfds5)]. A second reviewer was also initially unable to verify the same claims, was unsure if the theoretical improvements were sufficiently significant, and whether the method could apply to non-IF models of spiking neural networks [[2](https://openreview.net/forum?id=VQyHD2R3Aq&noteId=cc2tMpSst0t)]. The authors addressed this in their response by performing additional experiments with LIF neurons, but it wasn't clear if the main concerns were sufficiently addressed, since the reviewer did not change their score.

Based on the largely negative appraisal by all reviewers, I recommend that be paper be rejected. However, I strongly encourage the authors to revise and resubmit their paper to a future conference, focusing on making sure that their central claims can be more easily understood and verified.